# MC-Search: Evaluating and Enhancing Multimodal Agentic Search with Structured Long Reasoning Chains

**Xuying Ning**[1*], **Dongqi Fu**[2*], **Tianxin Wei**[1*], **Mengting Ai**[1], **Jiaru Zou**[1], **Ting-Wei Li**[1],
**Hanghang Tong**[1], **Yada Zhu**[3], **Hendrik Hamann**[4], **Jingrui He**[1†]
[1]University of Illinois Urbana-Champaign, [2]Meta, [3]IBM Research, [4]Stony Brook University
[*]Equal contribution. [†]Corresponding author.
{xuyingn2, jingrui}@illinois.edu
https://mc-search-project.github.io

## Abstract

With the increasing demand for step-wise, cross-modal, and knowledge-grounded reasoning, multimodal large language models (MLLMs) are evolving beyond the traditional fixed retrieve-then-generate paradigm toward more sophisticated agentic multimodal retrieval-augmented generation (MM-RAG). Existing benchmarks, however, mainly focus on simplified QA with short retrieval chains, leaving adaptive planning and multimodal reasoning underexplored. We present **MC-Search**, the first benchmark for agentic MM-RAG with long, step-wise annotated reasoning chains spanning five representative reasoning structures. Each example specifies sub-questions, retrieval modalities, supporting facts, and intermediate answers, with fidelity ensured by **HAVE** (Hop-wise Attribution and Verification of Evidence), resulting in 3,333 high-quality examples averaging 3.7 hops. Beyond answer accuracy, MC-Search introduces new process-level metrics for reasoning quality, stepwise retrieval and planning accuracy. By developing a unified agentic MM-RAG pipeline, we benchmark six leading MLLMs and reveal systematic issues such as over- and under-retrieval and modality-misaligned planning. Finally, we introduce **Search-Align**, a process-supervised fine-tuning framework leveraging verified reasoning chains, showing that our data not only enables faithful evaluation but also improves planning and retrieval fidelity in open-source MLLMs.

## 1 Introduction

Multimodal large language models (MLLMs) have rapidly advanced in their reasoning abilities (Wang et al., 2025b; Huang et al., 2025b; Li et al., 2024), moving beyond text-only inputs to interleaved cross-modal contexts. To ensure factuality and robustness, Retrieval-Augmented Generation (RAG) has emerged as a key paradigm for grounding model outputs in external textual evidence (Gao et al., 2023; Zhao et al., 2024; Fan et al., 2024). Building on this, Multimodal RAG (MM-RAG) equips MLLMs with *visual seeking* and extends *knowledge grounding* from text-only evidence to multimodal contexts, thereby facilitating *cross-modal reasoning* for knowledge-intensive queries.

In practice, queries for MLLMs are often ambiguous and complex (Figure 1). They go far beyond the shallow cases solvable with fixed minimal retrieval and instead require multi-step, cross-modal, and knowledge-intensive reasoning. As MLLMs advance toward long reasoning models (Wang et al., 2024), they are increasingly positioned as backbones for supporting such problem-solving needs. Realizing this potential, however, demands richer agentic behaviors (Li et al., 2025a;b), such as iterative task decomposition, adaptive cross-modal retrieval, and multimodal evidence integration. This shift calls for moving beyond the fixed retrieve-then-generate paradigm of classic RAG toward *multimodal agentic search-enhanced reasoning* (multimodal agentic RAG) (Wang et al., 2025a; Li et al., 2025a; Huang et al., 2025a; Jin et al., 2025; Zheng et al., 2025b; Chen et al., 2025). Accordingly, new benchmarks and evaluation pipelines are needed to faithfully capture and evaluate these capabilities.

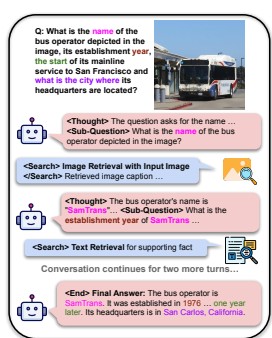

Figure 1: Example query requiring multimodal agentic search with a long reasoning chain.

Table 1: **Left**: Comparison of existing multimodal retrieval-augmented QA datasets. **Right**: Distribution of the five reasoning topologies in MC-SEARCH, with outer rings illustrating hop diversity (2–5 hops).

| Benchmark | Knowledge Modality | Typical Hops | Long Chain ($\geq 4$ Hops) | Stepwise Annotation | Reasoning Topology |
|---|---|---|---|---|---|
| OK-VQA (Marino et al., 2019) | Text | 1 Hop | ✗ | ✗ | ✗ |
| ViQuAE (Lerner et al., 2022) | Text | 1 Hop | ✗ | ✗ | ✗ |
| WebQA (Chang et al., 2022) | Text/Caption | ≤2 Hops | ✗ | ✗ | ✗ |
| InfoSeek (Chen et al., 2023) | Text | 1 Hop | ✗ | ✗ | ✗ |
| MMSearch (Jiang et al., 2024) | Text/Image | 1 Hop | ✗ | ✗ | ✗ |
| Dyn-VQA (Li et al., 2025b) | Text/Image | ≤2 Hops | ✗ | ✗ | ✗ |
| MRAG (Hu et al., 2025) | Image | 1 Hop | ✗ | ✗ | ✗ |
| M²RAG (Liu et al., 2025c) | Text/Image | 1 Hop | ✗ | ✗ | ✗ |
| **MC-SEARCH** | **Text/Image** | **≥4 Hops** | ✓ | ✓ | ✓ |

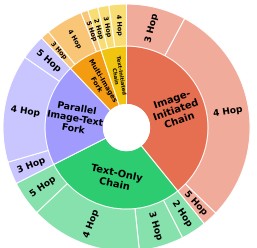

Although a number of recent benchmarks (Hu et al., 2025; Jiang et al., 2024; Liu et al., 2025c; Li et al., 2025b) have provided valuable evaluations of MM-RAG, they primarily focus on straightforward visual evidence: seeking questions with only 1–2 retrieval steps, thus falling short of examining the adaptive search-enhanced reasoning behaviors required in practice. Specifically, current datasets remain limited in three key respects: (i) most adopt simple question-answer formats with a fixed retrieve-then-generate pipeline that collapses multimodal evidence into a primary textual channel (Hu et al., 2025); (ii) they restrict evaluation to short 1–2 hop retrievals without long, adaptive reasoning trajectories (Li et al., 2025b); and (iii) they lack stepwise annotations and explicit reasoning topologies that clarify the roles of different modalities in the reasoning process (Hu et al., 2025; Liu et al., 2025b; Li et al., 2025b). As summarized in Table 1 (Left), these limitations make it difficult to determine whether MLLMs can truly perform long, structured reasoning over multimodal evidence.

To fill this gap, we present **MC-SEARCH**, the first benchmark for **M**ultimodal agentic RAG with long, structured **C**hains of **Search**-enhanced reasoning. Each example is paired with a golden step-wise trajectory specifying the sub-question sequence, retrieval modality, supporting fact, and intermediate answer, enabling fine-grained evaluation, laying the foundation for process reward modeling.

MC-SEARCH is designed to be **long**, **diverse**, and **non-redundant**. It covers five representative multi-hop reasoning structures as visualized in Figure 2, including *Text-Only Chain*, *Image-Initiated Chain*, *Text-Initiated Chain*, *Parallel Image-Text Fork*, and *Multi-Image Fork*, capturing both serial and parallel reasoning patterns across modalities. To guarantee that each hop is necessary and structurally meaningful, we introduce **HAVE** (**H**op-wise **A**ttribution and **V**erification of **E**vidence), which filters spurious or redundant steps, resulting in 3,333 high-quality annotated examples with an average chain length of 3.7 hops, surpassing existing benchmarks (Hu et al., 2025; Liu et al., 2025b; Li et al., 2025b). Table 1 (Right) reports the chain-length distribution across reasoning topologies.

Beyond answer-level accuracy, in MC-SEARCH, we propose three **process-level evaluation metrics**: (i) *LLM-as-a-Judge* for open-ended reasoning quality, (ii) *Structure-Aware per Step Hit Rate* for per-step retrieval fidelity, and (iii) *Rollout Deviation* to quantify execution drift. To evaluate the abilities, we further develop a unified agentic MM-RAG pipeline for fair benchmarking and extensive experiments with six leading MLLMs, including both proprietary and open-source models. Our analysis reveals key weaknesses such as over-retrieval, under-retrieval, and modality-specific planning errors, underscoring the need for better adaptive planning and retrieval over multimodal evidence.

In addition to serving as an evaluation resource, our data also provides training signals for improving model capabilities. To this end, we present **SEARCH-ALIGN**, a conversation-level alignment framework built on HAVE-verified reasoning chains. It applies supervised fine-tuning beyond final answers, using step-wise trajectories to provide process-level supervision that strengthens open-source models' ability to plan and retrieve across modalities.

Our main contributions are summarized as follows:

- **Benchmark:** We present MC-SEARCH, the first benchmark for agentic multimodal RAG with long, step-wise annotated reasoning chains spanning five representative structures, verified by the HAVE procedure for necessity and non-redundancy.

- **Metrics:** We propose new process-level metrics that move beyond answer accuracy to precisely attribute reasoning quality, per-step retrieval and planning fidelity.

- **Evaluation:** We develop a unified agentic MM-RAG pipeline and conduct extensive benchmarking of six leading MLLMs, revealing systematic issues such as over- and under-retrieval, modality-misaligned planning, and eight characteristic error types.

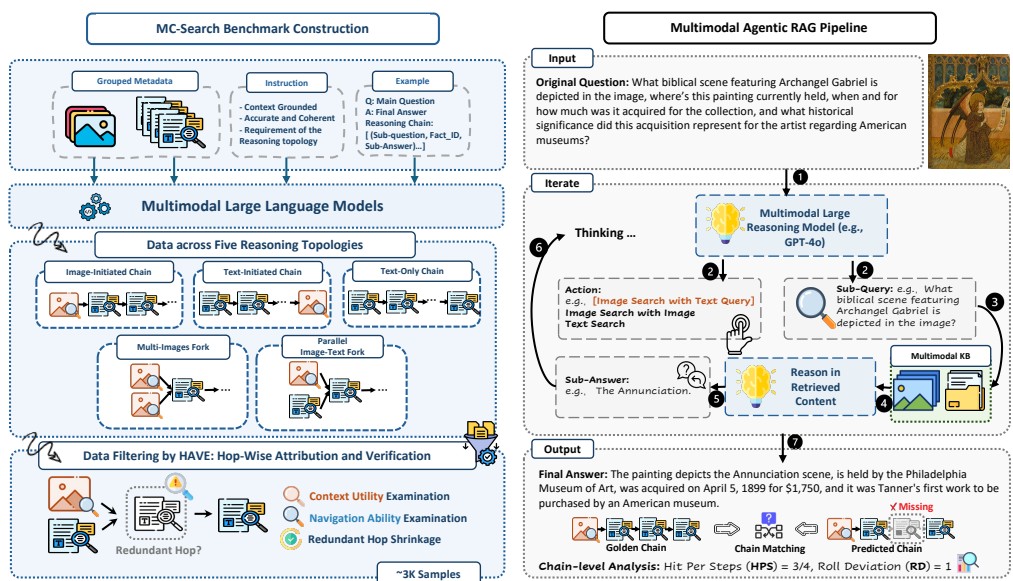

Figure 2: Overview of **MC-SEARCH** benchmark and evaluation. **Left**: Benchmark covering five reasoning topologies, filtered via the hop-wise attribution and verification of evidence (HAVE) process. **Right**: Multimodal agentic RAG pipeline, where an MLLM iteratively generates sub-queries and actions, retrieves multimodal evidence, reasons over the retrieved information, and integrates it to produce the final answer. Our framework further aligns predicted reasoning chains with golden trajectories to assess chain-level retrieval and planning.

- **Method:** We introduce SEARCH-ALIGN, which leverages verified reasoning chains for process-supervised fine-tuning of MLLMs, demonstrating the effectiveness of our data for training and improving planning and retrieval fidelity in agentic MM-RAG.

## 2 MC-SEARCH BENCHMARK

We propose MC-SEARCH, a large-scale benchmark for evaluating agentic multimodal RAG. In contrast to prior benchmarks that focus on shallow VQA tasks, MC-SEARCH provides long, step-wise verified reasoning chains across diverse reasoning topologies, enabling both fine-grained process-level analysis and model alignment. In this section, we detail the benchmark construction, dataset composition, and process-level evaluation metrics, with the overall workflow illustrated in Figure 2.

### 2.1 BENCHMARK CONSTRUCTION AND COMPOSITION

**Search-Enhanced Reasoning Topologies.** To capture the diversity of real-world agentic MM-RAG workflows, we design *five distinct reasoning topologies* that characterize how multimodal knowledge interacts and depends on each other within a long search-enhanced reasoning process. Each topology is represented as a reasoning graph (or reasoning chain) $\mathcal{G}(Q, A)$, associated with an overall question $Q$ and its final answer $A$. The graph consists of sub-questions $q_t$, retrieval modalities $m_t \in \{\text{text}, \text{image}\}$, retrieved evidence $r_t$, and intermediate answers $a_t$ derived from the evidence at hop $t$ (Figure 2). We denote by $\mathcal{R}$ the retrieval function that, given a sub-question $q_t$ and modality $m_t$, returns the corresponding evidence $r_t$. Formally,

$$\mathcal{G}(Q, A) = \{(q_t, m_t, r_t, a_t)\}_{t=1}^{T}, \quad r_t = \mathcal{R}(q_t, m_t), \quad A = f(\{a_t\}_{t=1}^{T}), \tag{1}$$

where $f(\cdot)$ denotes the reasoning procedure that aggregates intermediate answers into the final answer $A$ for the overall question $Q$.

Concretely, the five reasoning graphs are defined as: (i) *Image-Initiated Chain*, where reasoning is anchored in image retrieval at the first step and subsequently builds on textual retrieval, applies to both image-containing and text-only queries; (ii) *Text-Initiated Chain*, where reasoning begins with text retrieval of factual knowledge and later incorporates image retrieval for visual validation; (iii) *Parallel Image-Text Fork*, where image and text retrieval evolve concurrently without dependency across specific hops, so that interchanging the retrieval of such hop order does not alter the final answer, requiring cross-modal coordination; (iv) *Multi-Images Fork*, where multiple images must be

retrieved for visual comparison before turning to textual evidence for factual support; (v) *Text-Only Chain*, serving as a baseline where reasoning proceeds entirely through text. Data examples for each reasoning graph are provided in Appendix C.

**Data Generation.** We construct the MC-SEARCH dataset by collecting multimodal knowledge from Wikipedia and organizing it into coherent clusters of text and images, typically drawn from the same or closely related pages, following Chang et al. (2022). From these clusters, we curate salient text facts and associated images as the basis for multi-hop QA generation, supplemented with illustrative examples aligned with each reasoning topology. From the curated pools, we then generate questions at scale while ensuring balanced coverage across the five reasoning structures. Specifically, we prompt Gemini-2.5-Flash to select one topology and synthesize a structured multi-hop question together with its answer and a coherent reasoning chain grounded in supporting facts. This procedure yields approximately 21k examples aligned with the five search-enhanced reasoning topologies. Additional construction details and prompt templates are provided in Appendix N.

**Data Filtering.** A key challenge in constructing a high-quality dataset is that MLLMs often generate long reasoning chains with *hallucinated steps*, that is, plausible but ungrounded hops not supported by evidence, and *redundant steps* that do not contribute to the final answer. To address this, we propose **HAVE** (Hop-wise Attribution and Verification of Evidence), a data filtering mechanism that determines whether each hop in the reasoning chain is both *necessary* and *non-redundant*.

Let $\mathcal{C} = \{r_1, r_2, \ldots, r_T\}$ denote the set of retrieved evidence in each hop of a reasoning graph. We define the **context utility** of step $r_t$ as the drop in answer accuracy (measured by the F1 score between golden and generated answers) when $r_t$ is removed:

$$\text{Util}(t) = \text{F1}(\mathcal{C}) - \text{F1}(\mathcal{C} \setminus r_t). \tag{2}$$

A step is identified as necessary if $\text{Util}(t)$ exceeds a threshold. However, some steps may not directly affect answer accuracy but still serve to connect sub-questions and guide downstream reasoning. For example, in a two-hop chain, the first step may identify the building in the input image as "Christ Church Cathedral," while the second step queries factual knowledge about this entity. Although the first step has low context utility on its own, its extracted entity is essential for the second hop, making it indispensable for the reasoning process. To capture such cases, we further assess the **navigational role** of a step by checking whether entities in its intermediate answer $a_t$ appear in downstream sub-questions:

$$\text{Nav}(t) = \begin{cases} 1, & \text{if } \text{Ent}(a_t) \cap \text{Ent}(q_{t+1:T}) \neq \emptyset, \\ 0, & \text{otherwise,} \end{cases} \tag{3}$$

where $\text{Ent}(\cdot)$ denotes the set of entities extracted from a sub-question or an intermediate answer. If a step has $\text{Util}(t)$ below the threshold and $\text{Nav}(t) = 0$, it is marked as *redundant*.

We apply HAVE at scale by first filtering the generated dataset to remove samples with more than two redundant hops in their reasoning chains using the open-source Qwen2.5-VL-7B, yielding 4.8k candidate chains. To further mitigate model-specific bias, we then perform *hop shrinkage* and sample verification with Gemini-2.5-Pro, while simultaneously extracting key knowledge entities from the golden answers to facilitate future answer evaluation. Finally, we conduct a redundancy check using HAVE with the data-generation model Gemini-2.5-Flash, and yielding 3,333 high quality samples with all necessary hops and coherent reasoning trajectories. The final dataset composition and statistics are reported in Table 2 and Appendix E.

Table 2: Benchmark statistics.

| Statistic | Number |
|---|---|
| Parallel Image-Text Fork | 680 |
| Image–Initiated Chain | 1,306 |
| Text Chain | 945 |
| Text–Initiated Chain | 169 |
| Multi–Images Fork | 233 |
| **Total Samples** | **3,333** |
| KB Images | 389,750 |
| KB Documents | 784,473 |
| Avg. Hops | 3.79 |

Moreover, to ensure the reliability of subsequent step-wise retrieval accuracy evaluation, we enforce that the reasoning graph for each question is *unique*. To this end, we use Gemini-2.5-Flash for automatic verification, detecting and filtering confounding knowledge pieces in the knowledge base so that no unused text fact or image within the same topic cluster can independently resolve any sub-question. This provides a solid foundation for step-wise evaluation of agentic MM-RAG.

**Quality Verification.** To ensure dataset quality in terms of reasoning coherence, grounding, and QA accuracy, we employ Gemini-2.5-Pro to evaluate each chain on a 1–5 scale along four dimensions: factual correctness, step necessity, clarity, and multimodal alignment (Appendix F). The dataset

achieves an overall average score of 4.87, and low-scoring samples are further refined through targeted answer editing, yielding a high-quality benchmark for search-enhanced multimodal reasoning.

## 2.2 EVALUATION PROTOCOL

To evaluate multimodal search-enhanced reasoning, we believe that measuring only final answer accuracy is insufficient, especially for long reasoning trajectories where errors in retrieval or planning may not be reflected in the final output. To enable a more fine-grained assessment, we introduce metrics that capture both answer correctness and the fidelity of step-wise retrieval and planning.

**Answer Accuracy.** We use four complementary metrics: (i) **F1**, the standard token-level overlap score between the generated final answer and the golden answer; (ii) $\Delta$**F1**, which measures the performance gain brought by agentic MM-RAG beyond the model's parametric knowledge, defined as the difference between performance with and without retrieval: $\Delta$F1 = F1 $-$ F1$_{\text{w/o } \mathcal{R}}$; (iii) **Golden F1**, an upper bound obtained by supplying the model with gold reasoning chains and golden retrieval content; and (iv) **LLM-as-a-Judge (LJ)**, where a strong reasoning model (Gemini-2.5-Pro) evaluates predicted reasoning chains against the gold ones along four dimensions: answer accuracy, reasoning coherence, key knowledge entity coverage, and step alignment (prompt in Appendix P).

**Chain alignment.** We evaluate how well the predicted reasoning graph $\hat{\mathcal{G}}$ aligns with the golden reasoning graph $\mathcal{G}$. Each reasoning graph is a sequence of retrieval-augmented reasoning steps indexed by $t$, where $r_t \in \mathcal{G}$ denotes the evidence retrieved at golden step $t$, and $\hat{r}_{t'} \in \hat{\mathcal{G}}$ denotes the evidence retrieved at predicted step $t'$.

To assess step-level retrieval fidelity, we define the (i) **Hit per Step (HPS)** metric as the fraction of golden steps whose evidence is successfully recovered in the predicted graph:

$$\text{HPS}(\hat{\mathcal{G}}, \mathcal{G}) \;=\; \frac{1}{|\mathcal{G}|} \Big| \big\{ (t, t') \;\mid\; r_t \in \mathcal{G}, \; \hat{r}_{t'} \in \hat{\mathcal{G}}, \; \hat{r}_{t'} = r_t \big\} \Big|, \tag{4}$$

where $|\mathcal{G}|$ is the number of golden steps and $|\cdot|$ on the right is the number of matched steps. Each "hit" corresponds to a predicted step that exactly matches the evidence of a golden step, with duplicates counted only once. Beyond exact matching, we further align $\hat{\mathcal{G}}$ and $\mathcal{G}$ through *maximum-weight bipartite matching*, where nodes correspond to steps in the two graphs and edge weights reflect evidence similarity between predicted and golden steps. This alignment enables fine-grained step-wise comparison and better handles cases involving parallel reasoning steps.

To capture structural fidelity, we measure the step-length gap between the predicted and golden reasoning graphs. A large gap indicates under- or over-retrieval, while a small value suggests closer alignment in reasoning complexity. We refer to this metric as (ii) **Rollout Deviation (RD)**:

$$\text{RD}(\hat{\mathcal{G}}, \mathcal{G}) = \big| \, |\hat{\mathcal{G}}| - |\mathcal{G}| \, \big|. \tag{5}$$

## 3 AGENTIC MM-RAG PIPELINE AND PROCESS-LEVEL ALIGNMENT

To systematically evaluate and improve MLLMs in search-enhanced reasoning, we introduce a unified agentic MM-RAG pipeline for adaptive planning, retrieval for multimodal evidence, and reasoning over evidence, together with SEARCH-ALIGN, a process-level alignment framework that leverages verified reasoning chains to strengthen open-source models.

### 3.1 AGENTIC MM-RAG PIPELINE

As illustrated in Figure 2 (Right), our pipeline models multimodal search-enhanced reasoning as an iterative process with three modules: (i) *sub-query and action generation*, (ii) *evidence acquisition*, and (iii) *iterative reasoning and synthesis*. At each iteration, the agent formulates a sub-goal, selects a modality-aware retrieval action, integrates the retrieved evidence, and decides whether to continue searching or terminate with a final answer. This design supports not only effective reasoning but also fine-grained, step-level evaluation for agentic RAG.

**(i) Sub-query and Action Generation.** Given a complex question, the MLLM first predicts the next sub-goal and generates a focused sub-query (steps 1–2 in Figure 2). It then adaptively chooses one of three retrieval actions, corresponding to different ways of accessing multimodal evidence: (a) text search with a text query, (b) image search with a text query, or (c) image search with an input image.

**(ii) Evidence Acquisition.** The selected action is executed over our local multimodal knowledge base. The sub-query is embedded by a multimodal encoder for dense retrieval in the corresponding modality, returning textual or visual evidence. For each sub-query, we keep the top-1 evidence by query-answer similarity (step 3) and let the MLLM generate a sub-answer (steps 4–5) after reasoning. Both the chosen modality and retrieved evidence are logged for process evaluation.

**(iii) Iterative Reasoning and Synthesis.** The sub-answer and its evidence are fed back into the model to guide the next planning step (step 6), forming a loop of planning, retrieval, and reasoning. The agent dynamically checks whether the accumulated evidence is sufficient; if so, it outputs a final response (step 7). Otherwise, it continues with another sub-query. This iterative design exposes the full reasoning trajectory, enabling modality-aware execution and precise chain-level evaluation.

## 3.2 SEARCH-ALIGN: PROCESS-LEVEL ALIGNMENT

Beyond evaluation, our dataset also serves as a resource for improving model capabilities. We introduce **SEARCH-ALIGN**, a conversation-level alignment framework for open-source MLLMs built on HAVE-verified reasoning chains. Unlike conventional SFT that supervises only final answers, SEARCH-ALIGN provides process-level supervision by leveraging step-wise annotated trajectories with sub-questions, retrieval actions, evidence, and intermediate answers.

**Training Data Construction.** Each reasoning graph is augmented with explicit reasoning thoughts generated by Gemini-2.5-Flash, which explain how to ground reasoning in evidence and connect adjacent hops. This converts step-wise annotations into coherent dialogue traces: the assistant poses sub-questions and reasons over evidence, while the user executes retrieval actions and returns results.

**Supervised Fine-Tuning (SFT).** We then fine-tune MLLMs on these conversation-style traces, enabling them to learn not only to produce correct answers but also to plan, choose retrieval modalities, and integrate evidence across steps. This process-level alignment provides a richer training signal that better prepares models for long-horizon multimodal reasoning.

## 4 EXPERIMENTS

In this section, we evaluate leading MLLMs on the MC-SEARCH benchmark under our unified agentic MM-RAG pipeline, focusing on five research questions (RQs): **RQ1**: How do MLLMs perform across reasoning structures, and does SEARCH-ALIGN improve open-source models? **RQ2**: How does reasoning chain length affect reasoning performance? **RQ3**: What are the effects of over- and under-retrieval? **RQ4**: Do models show inherent modality preferences? **RQ5**: What are the typical error types in multimodal agentic reasoning?

## 4.1 EXPERIMENTAL SETUP

We evaluate the following MLLMs using the agentic MM-RAG pipeline introduced in Section 3.1:

- **Closed-source models**: GPT-4o-Mini (Achiam et al., 2023), Gemini-2.5-Flash (Team et al., 2023), Gemini-2.5-Pro (Team et al., 2023), and Claude-3.7-Sonnet (Anthropic, 2024), representing the strongest commercially available reasoning-oriented MLLMs.
- **Open-source models**: InternVL3.5-8B (Wang et al., 2025b) and Qwen2.5-VL-7B (Bai et al., 2025), two state-of-the-art vision-language models. We also evaluate their performance after using SEARCH-ALIGN framework.

**Evaluation protocol.** All models are evaluated under identical conditions. We use the same embeding model for multimodal retrieval (Zhou et al., 2024), prompts (Appendix O), and decoding parameters across models. Each retrieval step is restricted to the top-1 result from the local multimodal knowledge base, and the same maximum number of reasoning iterations is enforced. Performance is reported using the six metrics in Section 2.2: answer accuracy (F1, $\Delta$F1, Golden F1, and LLM-as-a-Judge) and chain-level retrieval and planning fidelity (Hit per Step and Rollout Deviation). Additional details on evaluation and the training of SEARCH-ALIGN are provided in Appendix I. In the Appendix D.1, we additionally report fixed-step RAG baselines (1-hop and 2-hop) for the backbone models to complement the main results.

Table 3: Evaluation of MLLMs on the MC-SEARCH benchmark under the agentic MM-RAG pipeline, reported across five reasoning graphs. Best backbone results are shown in **bold**, second-best are underlined, and improvements from SEARCH-ALIGN on open-source models are highlighted in **bold red**.

| Reasoning $\mathcal{G}$ | | Model | Answer Accuracy | | | Chain Alignment | | Golden F1 (↑) |
|---|---|---|---|---|---|---|---|---|
| | | | F1 (↑) | ΔF1 (↑) | LJ (↑) | HPS (↑) | RD (↓) | |
| **Image-Initiated Chain** | CLOSE | GPT-4o-Mini | 36.49 | 34.18 | 2.63 | 27.51 | 1.46 | 68.29 |
| | | Gemini-2.5-Flash | 44.10 | 37.38 | 3.01 | **31.46** | 2.91 | 72.39 |
| | | Gemini-2.5-Pro | **47.61** | **42.76** | **3.18** | 25.90 | **1.05** | 69.83 |
| | | Claude-3.7-Sonnet | 37.80 | 33.09 | 2.60 | 27.31 | 1.18 | **72.62** |
| | OPEN | InternVL3.5-8B | 39.11 | 29.49 | 2.27 | 22.59 | 1.58 | 63.86 |
| | | + SEARCH-ALIGN | **42.27** | **32.65** | **2.53** | **32.49** | **0.94** | |
| | | Qwen2.5-VL-7B | 26.30 | 8.65 | 1.34 | 16.51 | 4.04 | 60.95 |
| | | + SEARCH-ALIGN | **45.70** | **28.05** | **2.23** | **33.59** | **0.70** | |
| **Multi-Images Fork** | CLOSE | GPT-4o-Mini | 24.00 | 19.16 | 2.13 | 16.04 | 1.94 | 59.46 |
| | | Gemini-2.5-Flash | 36.80 | 31.89 | 2.35 | 13.45 | 1.66 | **64.40** |
| | | Gemini-2.5-Pro | **40.37** | **36.58** | **2.76** | 18.68 | **1.40** | 61.29 |
| | | Claude-3.7-Sonnet | 29.78 | 27.85 | 2.07 | **19.43** | 1.45 | 62.41 |
| | OPEN | InternVL3.5-8B | 33.08 | 20.98 | 2.13 | 16.02 | 2.25 | 57.65 |
| | | + SEARCH-ALIGN | **36.53** | **24.43** | **2.49** | **39.33** | **1.67** | |
| | | Qwen2.5-VL-7B | 26.13 | 7.53 | 1.41 | 12.58 | 3.80 | 57.04 |
| | | + SEARCH-ALIGN | **38.01** | **19.41** | **2.11** | **38.01** | **1.16** | |
| **Text-Initiated Chain** | CLOSE | GPT-4o-Mini | 30.11 | 18.46 | 2.41 | **27.18** | 1.76 | 49.47 |
| | | Gemini-2.5-Flash | 43.55 | 26.34 | 3.30 | 25.20 | 1.20 | **66.27** |
| | | Gemini-2.5-Pro | **45.30** | **29.89** | **3.62** | 19.51 | **0.95** | 55.94 |
| | | Claude-3.7-Sonnet | 36.10 | 17.36 | 2.63 | 25.09 | 1.12 | 57.13 |
| | OPEN | InternVL3.5-8B | 38.34 | 25.48 | **2.60** | 19.06 | 1.97 | 38.70 |
| | | + SEARCH-ALIGN | **40.87** | **28.01** | 2.59 | **37.82** | **1.44** | |
| | | Qwen2.5-VL-7B | 31.47 | 14.34 | 1.66 | 17.02 | 3.80 | 39.86 |
| | | + SEARCH-ALIGN | **45.07** | **27.94** | **2.42** | **33.59** | **0.87** | |
| **Parallel Image-Text Fork** | CLOSE | GPT-4o-Mini | 21.98 | 21.65 | 2.20 | 15.00 | 1.71 | 53.98 |
| | | Gemini-2.5-Flash | 29.92 | 27.94 | 2.58 | 11.43 | 2.70 | 57.99 |
| | | Gemini-2.5-Pro | **34.83** | **34.19** | **2.99** | **16.74** | **1.29** | 53.46 |
| | | Claude-3.7-Sonnet | 26.43 | 24.65 | 1.95 | 14.32 | 1.37 | 58.51 |
| | OPEN | InternVL3.5-8B | 29.74 | 23.05 | 2.13 | 14.22 | 1.75 | **58.82** |
| | | + SEARCH-ALIGN | **30.72** | **24.03** | **2.29** | **23.55** | **1.45** | |
| | | Qwen2.5-VL-7B | 22.94 | 14.32 | 1.33 | 10.09 | 3.93 | 55.76 |
| | | + SEARCH-ALIGN | **32.73** | **24.11** | **1.78** | **25.07** | **1.05** | |
| **Text-Only Chain** | CLOSE | GPT-4o-Mini | 30.96 | 30.94 | 2.58 | 21.94 | 1.13 | 61.90 |
| | | Gemini-2.5-Flash | 34.03 | 33.96 | 2.49 | 20.59 | **1.04** | **67.72** |
| | | Gemini-2.5-Pro | 34.47 | **34.46** | **2.66** | 21.59 | 1.07 | 62.42 |
| | | Claude-3.7-Sonnet | 27.37 | 26.75 | 2.08 | **29.64** | 3.14 | 66.38 |
| | OPEN | InternVL3.5-8B | **34.82** | 27.84 | 2.45 | **21.81** | 2.15 | 65.72 |
| | | + SEARCH-ALIGN | **38.45** | **31.47** | **2.68** | 20.54 | **1.28** | |
| | | Qwen2.5-VL-7B | 23.54 | 14.89 | 1.36 | 13.35 | 4.63 | 61.57 |
| | | + SEARCH-ALIGN | **37.54** | **28.89** | **1.99** | **19.04** | **0.98** | |

## 4.2 MAIN RESULTS (RQ1)

**Backbone Performance and Search-Align Gains.** As shown in Table 3, proprietary models such as Gemini-2.5-Pro achieve the highest overall accuracy across most reasoning topologies. Among open-source models, InternVL3.5-8B is the stronger backbone, while Qwen2.5-VL-7B consistently lags behind. With SEARCH-ALIGN, however, both backbones see substantial gains: InternVL improves by about +2.8 F1 and +12.0 HPS on average while reducing RD by 0.6, and QwenVL achieves even larger boosts of +13.7 F1 and +16.0 HPS with a –3.1 drop in RD. After alignment, Qwen2.5-VL-7B nearly matches Gemini-2.5-Pro on text-centric chains, and InternVL shows stable improvements across all topologies. These results indicate that a major weakness of open-source models lies in retrieval planning and step-level reasoning alignment rather than basic perceptual or semantic understanding, and that these capabilities can be effectively improved through training on high-quality search-enhanced reasoning data.

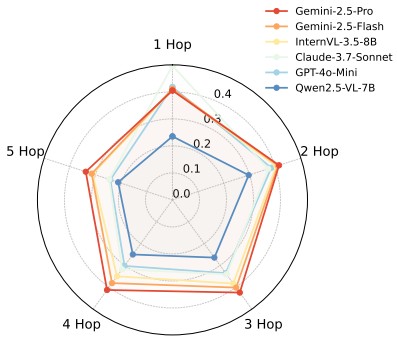 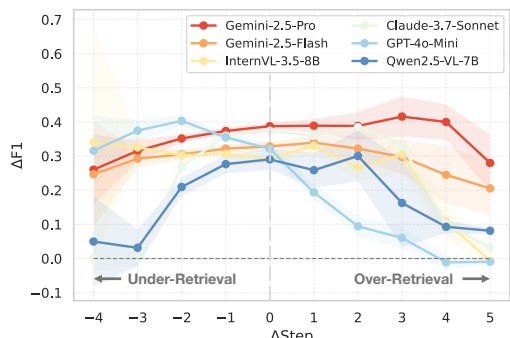

Figure 3: Model F1 across samples with different chain lengths, showing a consistent drop in performance as the chain length increases.

Figure 4: Model $\Delta$F1 vs. $\Delta$ Step (difference in length between generated and golden reasoning chains), where larger positive $\Delta$steps indicate more over-retrieval.

**Topology-Specific Challenges.** Across reasoning topologies, we observe *Parallel Image-Text Fork* is the most challenging, where all backbones reach their lowest F1 and HPS. The difficulty comes from the need to plan and cover both text and image branches simultaneously, so missing either branch directly lowers HPS, and under the top-1 retrieval constraint models often fail to capture the full evidence set. Aligning parallel evidence without strict order further increases the risk of incomplete or inconsistent reasoning. *Multi-Images Fork* is also difficult, mainly due to the challenge of aligning multiple visual evidences with corresponding textual facts, where models often retrieve the right images but misattribute or fail to integrate them coherently. In contrast, *Text-Initiated* and *Text-Only Chains* are relatively easier, as they rely primarily on text retrieval and linear reasoning, which align with model strengths. *Image-Initiated Chains* are also tractable: once the initial visual step is correct, subsequent textual validation is linear and more stable than multi-branch reasoning.

## 4.3 ANALYSIS

**RQ2: Performance vs. Chain Length.** Figure 3 reports model F1 across samples with different chain lengths, showing a consistent drop in performance as the chain length increases. Models remain relatively robust on short reasoning chains (1–3 hops), where performance differences across backbones are moderate, with the exception of Qwen2.5-VL-7B, which lags behind consistently even at early hops. When the chain length extends to 4 or 5 hops, however, all models experience a sharp degradation, reflecting the compounding difficulty of sustaining long-horizon reasoning.

Among closed-source systems, Gemini-2.5-Pro is the most robust, sustaining the highest performance at longer hops. Gemini-2.5-Flash and InternVL3.5-8B degrade gradually, while GPT-4o-Mini and Claude-3.7-Sonnet drop sharply beyond 3 hops. Qwen2.5-VL-7B remains weakest across all lengths. Overall, the main challenge lies in extended multi-step chains, where compounding retrieval errors and unstable planning reduce accuracy.

**RQ3: Over- and Under-Retrieval Analysis.** Figure 4 shows $\Delta$F1 across different $\Delta$Steps, where negative values correspond to under-retrieval and positive values indicate over-retrieval. The results reveal that both extremes are harmful, though in different ways. When models severely under-retrieve ($\Delta$Step $< -2$), performance drops sharply as essential evidence is skipped, creating gaps that later reasoning cannot recover. Conversely, moderate over-retrieval ($\Delta$Step $= 1$–2) often improves accuracy: Gemini-2.5-Pro, Gemini-2.5-Flash, and InternVL3.5-8B achieve their highest $\Delta$F1 here, suggesting that one or two extra turns can help recover from imperfect planning.

However, when over-retrieval is excessive ($\Delta$Step $\geq 4$), noise and irrelevant context lead to sharp drops across all models. Gemini-2.5-Pro is most resilient, Gemini-2.5-Flash and InternVL3.5-8B remain moderately stable, while Claude-3.7-Sonnet and GPT-4o-Mini are highly vulnerable. This underscores a central tension between capturing sufficient evidence and avoiding over-retrieval.

**RQ4: Modality Coverage Analysis.** As shown in Table 4, which reports step-level retrieval coverage (covered vs. gold steps) for both image and text modalities under queries with and without image input, both models maintain strong text coverage ($>78\%$) across all settings, but image coverage is far weaker and highly dependent on explicit image inputs. For example, Gemini-2.5-Pro drops from 87.35% with image mentions to 29.50% without, while InternVL3.5-8B falls from 63.84% to 0.66%.

These findings expose a pronounced modality gap, as models tend to default to text retrieval and lose visual grounding in the absence of explicit image cues.

Table 4: Stepwise retrieval modality coverage (Cov.). Red = high coverage; Blue = low coverage.

| Type | Modality | Gemini-2.5-Pro | | InternVL-3.5-8B | |
|---|---|---|---|---|---|
| | | Covered/Gold | Cov. (%) | Covered/Gold | Cov. (%) |
| Query w/ Image | Image | 1,339/1,533 | 87.35 | 977/1,533 | 63.84 |
| | Text | 3,230/4,109 | 78.61 | 3,395/4,109 | 82.67 |
| Query w/o Image | Image | 300/1,017 | 29.50 | 7/1,017 | 0.66 |
| | Text | 2,097/2,510 | 83.55 | 2,251/2,510 | 89.78 |
| Overall | Image | 1,639/2,550 | 64.27 | 861/2,550 | 33.69 |
| | Text | 5,327/6,619 | 80.48 | 5,688/6,619 | 85.96 |

**RQ5: Error Analysis.** Figure 5 shows the distribution of eight error types, annotated using Gemini-2.5-Pro as the judge. The most frequent errors are *Retrieval-Failure* (84.7%), *Hallucinated-Entity/Attribute* (75.8%), and *Step-Omission* (74.3%), revealing that current models often fail to ground reasoning in relevant evidence, introduce unsupported content, or skip key steps.

Mid-frequency errors (e.g., modality mismatch, spurious steps) reflect unstable planning, while structural errors (order/dependency mistakes, multi-hop failures) are rarer but, given limited retrieval, remain non-trivial for future study. Evidence misinterpretation is infrequent, suggesting models can handle retrieved context; major failures arise earlier, during planning and evidence acquisition. Overall, these patterns, together with our HPS and over/under-retrieval analyses, identify retrieval fidelity (modality-aware retrieval, sufficient planning/hop coverage, and calibrated stopping) as the central bottleneck for reliable chain-level reasoning.

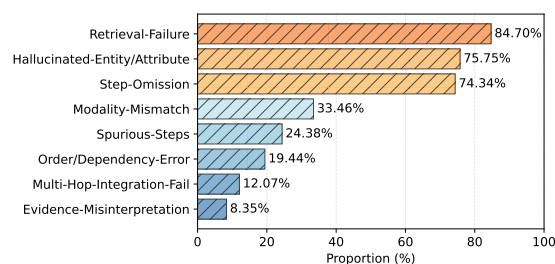

Figure 5: Eight-way error taxonomy proportions (higher is worse), annotated by Gemini-2.5-Pro.

## 5 RELATED WORK

We review existing work on MM-RAG benchmarks and Agentic RAG, with full details in Appendix B.

**Multimodal RAG Benchmarks.** MM-RAG extends retrieval-augmented generation by incorporating image evidence for cross-modal reasoning (Xia et al., 2024; Chang et al., 2022). Existing benchmarks are limited: most use fixed retrieve-then-generate pipelines (Marino et al., 2019; Hu et al., 2025), reduce images to captions (Lerner et al., 2022; Chen et al., 2023), restrict reasoning to 1–2 hops (Li et al., 2025b; Liu et al., 2025c), and lack step-wise annotations, making it hard to assess multimodal search-enhanced reasoning. We introduce MC-SEARCH, with HAVE-verified long reasoning trajectories across diverse topologies, enabling fine-grained attribution and analysis.

**Agentic RAG.** Recent work reframes retrieval as sequential decision-making, where agentic RAG systems decompose queries, adaptively retrieve, and synthesize knowledge (Wang et al., 2025a; Li et al., 2025a; Zheng et al., 2025b). While effective, most remain text-only and focus on multihop QA or scientific domains (Trivedi et al., 2022; Rein et al., 2024). Our benchmark fills this gap by extending agentic RAG to multimodal settings with diverse, and step-wise verified reasoning chains.

## 6 CONCLUSION

We present MC-SEARCH, a benchmark for structured, step-wise multimodal retrieval-augmented reasoning, encompassing five diverse reasoning mechanisms. With fine-grained annotations, hop-wise attribution and verification, and new chain-level metrics, MC-SEARCH enables thorough diagnosis of retrieval planning and reasoning quality. Our analyses highlight the importance of adaptive, modality-aware search strategies in agentic MM-RAG systems. Looking ahead, we envision MC-SEARCH as a foundation for advancing multimodal agents and fostering principled evaluation standards for agentic reasoning. we will broaden evaluations to stronger reasoning models and extend the benchmark to additional domains such as science and mathematics.

ACKNOWLEDGEMENTS

This work is supported by NSF (2134079), and IBM-Illinois Discovery Accelerator Institute. The content of the information in this document does not necessarily reflect the position or the policy of the Government, and no official endorsement should be inferred. The U.S. Government is authorized to reproduce and distribute reprints for Government purposes notwithstanding any copyright notation here on.

**Ethics Statement.** Our benchmark is constructed entirely from publicly available sources (e.g., Wikipedia text and images), without involving private data, or sensitive content. We apply a hop-wise attribution and verification (HAVE) process to filter hallucinated or redundant steps, ensuring factual grounding and minimizing risks of misleading information. The dataset is released solely for research purposes to advance multimodal reasoning models and agentic RAG, and not intended for sensitive or high-stakes applications.

**Reproducibility Statement.** We document the benchmark construction pipeline, filtering process, evaluation protocols, and experimental setups in detail within the paper and appendix. To facilitate full reproducibility, we have publicly released the dataset and code after acceptance, enabling independent verification and replication of all reported results.

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

# Appendix

# Contents

## A    USE OF LARGE LANGUAGE MODELS

During the preparation of this paper, we made controlled use of LLMs, specifically ChatGPT, as an auxiliary writing tool. The LLM was employed solely for stylistic refinement, namely to improve the fluency, grammar, and readability of paragraphs that were originally drafted by the authors. Importantly, the scientific content, methodology, experimental design, and main narrative of the paper were fully conceived, written, and validated by the authors without reliance on LLMs. Therefore, LLMs served purely in a supportive role for polishing author-written text, and their contribution does not rise to the level of co-authorship.

## B    RELATED WORK

We organize the related work from three complementary perspectives: (1) multimodal RAG benchmarks that extend retrieval beyond text, (2) agentic RAG systems that model retrieval as a sequential decision process, and (3) structure-aware benchmarks.

### B.1    MULTIMODAL RAG BENCHMARKS

MM-RAG extends retrieval-augmented generation beyond text by incorporating image evidence for knowledge-intensive, cross-modal reasoning (Xia et al., 2024; Chang et al., 2022; Pan et al., 2024). Existing benchmarks offer valuable first steps but remain limited: most adopt simple QA formats with fixed retrieve-then-generate pipelines (Marino et al., 2019; Hu et al., 2025; Chang et al., 2022; Amirshahi et al., 2025; Friel et al., 2024), perform limited visual seeking that reduces image information to captions (Chang et al., 2022; Lerner et al., 2022; Chen et al., 2023), restrict reasoning to 1–2-hop retrievals in VQA-style tasks (Li et al., 2025b; Liu et al., 2025c), and lack step-wise annotations or explicit reasoning structures. These constraints hinder diagnosing whether MLLMs genuinely perform structured multimodal reasoning or merely exploit shortcuts. To address this, We introduce MC-SEARCH, which provides HAVE-verified long trajectories across diverse reasoning topologies enabling fine-grained attribution and comprehensive analysis of MLLMs' long-horizon, search-enhanced reasoning.

### B.2    AGENTIC SEARCH-ENHANCED REASONING

Recent work reframes retrieval as a sequential decision process, giving rise to agentic RAG systems that decompose queries, adaptively retrieve evidence, and synthesize knowledge from retrieved contexts (Wang et al., 2025a; Li et al., 2025a; Zheng et al., 2025b; Singh et al., 2025; Pan et al., 2024). Supervised fine-tuning (Li et al., 2025a) and reinforcement learning methods (Chen et al., 2025; Jin et al., 2025) further align model behavior with ground-truth reasoning traces to enhance interpretability and grounding fidelity. However, these advances remain largely text-only, often targeting multihop QA (Trivedi et al., 2022; Yang et al., 2018) or scientific domains (Rein et al., 2024), and thus lack systematic study of how multimodal signals affect retrieval planning and step-wise supervision in MLLMs, and how different search-enhanced reasoning topologies influence model behavior. Our work addresses this gap by introducing a multimodal benchmark with diverse search-enhanced reasoning structures, step-wise verified long reasoning chains, and process-level metrics, revealing challenges such as modality-misaligned planning, under- and over-retrieval, and error taxonomy, and by proposing SEARCH-ALIGN to improve planning and retrieval fidelity through step-wise supervision.

### B.3    STRUCTURE-AWARE BENCHMARKS

A related line of work investigates reasoning models operating over symbolic KBs or graphs, often with schema-constrained queries or specialized tools (Lee et al.; Shen et al., 2024; Zhang et al., 2025; Wang & Pan, 2022; Feng et al., 2020; Liu et al., 2025a). These improve symbolic grounding but mainly focus on *what* to retrieve. In contrast, benchmarking agentic MM-RAG requires supervision over the *structure of the retrieval chain* itself—covering initiation modality, hop order, and parallel vs. serial patterns. To our knowledge, no prior multimodal dataset provides such structured, step-wise annotations. Our Search-MM fills this gap by supplying golden reasoning chains with modality-specific retrieval steps, enabling fine-grained attribution, analysis of over-/under-retrieval, and evaluation of diverse multimodal reasoning structures.

## C   DATA EXAMPLE

### C.1   IMAGE-INITIATED CHAIN

---

**Example Reasoning Chain — Image-Initiated Chain**

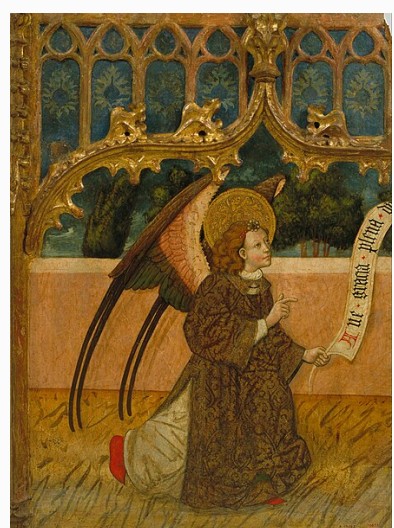

*Caption:* Archangel Gabriel from an Annunciation - Google Art Project

**Main Question.** What biblical scene featuring Archangel Gabriel is depicted in the image, where is this painting currently held, when and for how much was it acquired for the collection, and what historical significance did this acquisition represent for the artist regarding American museums?

**Final Answer.** The painting depicts the *Annunciation* scene, is held by the *Philadelphia Museum of Art*, was acquired on *April 5, 1899* for *$1,750*, and it was *Tanner's first work to be purchased by an American museum*.

**Reasoning Chain (4 steps):**

1. **(Image)** What biblical scene featuring Archangel Gabriel is depicted in the image?
   *Image Evidence ID:* `30332773`
   *Answer:* The Annunciation.

2. **(Text)** Which museum holds this painting?
   *Text Evidence ID:*
   `d5be2ae00dba11ecb1e81171463288e9_0`
   *Answer:* Philadelphia Museum of Art.

3. **(Text)** When and for what price was the painting acquired for the collection?
   *Text Evidence ID:*
   `d5be2ae00dba11ecb1e81171463288e9_1`
   *Answer:* April 5, 1899 for $1,750.

4. **(Text)** What was the historical significance of this acquisition for the artist regarding American museums?
   *Text Evidence ID:*
   `d5be2ae00dba11ecb1e81171463288e9_8`
   *Answer:* It was Tanner's first work to be purchased by an American museum.

---

## C.2 TEXT-INITIATED CHAIN

**Example Reasoning Chain — Text-Initiated Chain**

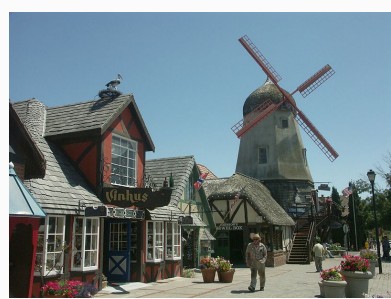

*Caption:* Solvang California Windmill.

**Main Question.** Does the image visually confirm Solvang's unique cultural character as described in the text?

**Final Answer.** *Yes*, the image prominently features a large windmill and buildings with distinct European/Danish architectural styles, aligning with the description of Solvang as "a bit of Denmark" in California.

**Reasoning Chain (2 steps):**

1. **(Text)** How is Solvang culturally characterized according to the text?
   *Text Evidence ID:* `d5bd142a0dba11ecb1e81171463288e9_7`
   *Answer:* Solvang is described as "a bit of Denmark tucked into this beautiful California."

2. **(Image)** Does the image visually confirm this cultural characterization, and if so, how?
   *Image Evidence ID:* `30326437`
   *Answer:* Yes, the image prominently displays a large windmill and buildings with distinct European/Danish architectural styles, confirming its cultural description.

## C.3 TEXT CHAIN

**Example Reasoning Chain — Text Chain**

**Main Question.** What are the key attributes of the Bentley Mulsanne, including its manufacturing period, engine specifications, origin of its name, and notable special editions?

**Final Answer.** The Bentley Mulsanne, a full-size luxury car *manufactured from 2010 to 2020*, is named after the Mulsanne Corner of the Le Mans racing circuit. It uses a *6.75 L Bentley L Series V8 engine*, and a notable special edition is the *"W.O. Edition"*, which features a piece of W.O. Bentley's personal car crankshaft.

**Reasoning Chain (3 steps):**

1. **(Text)** What type of vehicle is the Bentley Mulsanne, when was it manufactured, and what is the origin of its name?
   *Text Evidence ID:* `d5bd6ace0dba11ecb1e81171463288e9_15`
   *Answer:* The Bentley Mulsanne is a full-size luxury car that was manufactured from 2010 to 2020 and is named after the Mulsanne Corner of the Le Mans racing circuit.

2. **(Text)** What are the engine specifications of the Bentley Mulsanne?
   *Text Evidence ID:* `d5bd6ace0dba11ecb1e81171463288e9_6`
   *Answer:* The Mulsanne uses a 6.75 L (6,750 cc/411 in³) Bentley L Series V8 engine, modified to meet Euro V emissions regulations.

3. **(Text)** What notable special edition of the Mulsanne was introduced and what was its unique feature?
   *Text Evidence ID:* `d5bd6ace0dba11ecb1e81171463288e9_11`
   *Answer:* The Mulsanne "W.O. Edition" was presented, featuring a piece of the crankshaft from W.O. Bentley's personal car displayed in the arm rest.

## C.4 PARALLEL VISUAL-TEXTUAL FORK

---

**Example Reasoning Chain — Parallel Visual-Textual Fork**

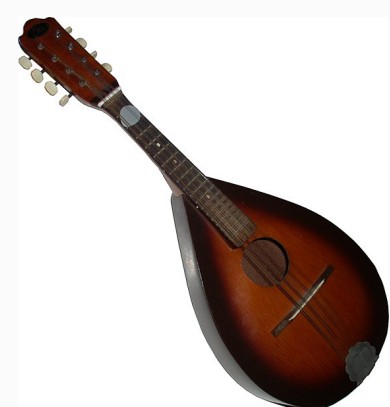

*Caption:* Mandolin1.

**Main Question.** Given the appearance of the mandolin shown, how did Pasquale Vinaccia's 1835 innovation involving string material lead to structural changes in the instrument and its lasting impact on mandolin design?

**Final Answer.** Pasquale Vinaccia's 1835 innovation of using *steel wire strings*, which are visible on the mandolin, necessitated *strengthening the body* and *deepening the bowl* for increased *resonance*, and this method of stringing ultimately *became the dominant way* for mandolins.

**Reasoning Chain (4 steps):**

1. **(Image)** What type of strings are visible on the mandolin depicted, and how does this visual detail relate to Pasquale Vinaccia's historical improvements to the instrument?
   *Image Evidence ID:* `30204742`
   *Answer:* The mandolin prominently displays wire/steel strings."

2. **(Text)** What structural modifications were subsequently required for the mandolin's body due to the adoption of these new wire strings?
   *Text Evidence ID:* `d5be68020dba11ecb1e81171463288e9_15`
   *Answer:* The wire strings necessitated strengthening the body and deepening the bowl.

3. **(Text)** What specific acoustic quality was enhanced by deepening the mandolin's bowl as a consequence of these structural changes?
   *Text Evidence ID:* `d5be68020dba11ecb1e81171463288e9_3`
   *Answer:* Deepening the bowl increased tonal resonance.

4. **(Text)** Considering these improvements, what was the long-term impact of Vinaccia's decision to use steel strings on mandolin design and construction?
   *Text Evidence ID:* `d5be68020dba11ecb1e81171463288e9_5`
   *Answer:* His steel-stringing approach became the dominant way of stringing mandolins.

---

## C.5 Multi-Images Fork

**Example Reasoning Chain — Multi-Images Fork**

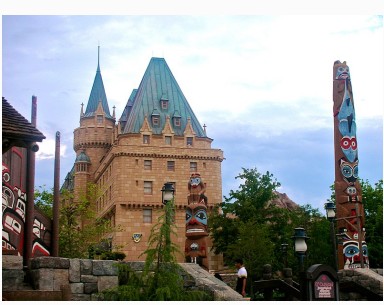

*Caption:* Canada Pavilion

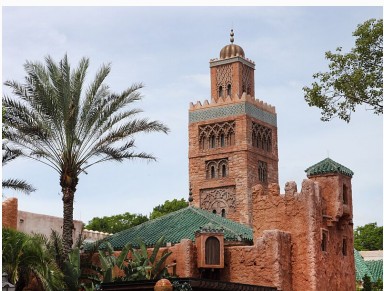

*Caption:* Morocco Pavilion

**Main Question.** Which of the depicted Epcot pavilions, the Canada or Morocco, features a prominent tall, slender tower as its main architectural centerpiece, and what is its historical significance regarding its addition to the World Showcase?

**Final Answer.** The Morocco Pavilion features *a prominent tall, slender tower*, and it was historically significant as *the first expansion pavilion added to World Showcase*, opening on September 7, 1984.

**Reasoning Chain (4 steps):**

1. **(Image)** What is the most prominent feature of the roof structure shown on the main building, the Canada Pavilion?
   *Image Evidence ID:* 30383576
   *Answer:* A very steep, multi-tiered green roof with pointed spires.

2. **(Image)** What is the most prominent architectural feature defining the skyline of the building in the Morocco Pavilion?
   *Image Evidence ID:* 30021436
   *Answer:* A tall, rectangular tower topped with a small dome.

3. **(Image)** Considering the contrast in primary vertical architectural elements between the Canada Pavilion's steep roof and the Morocco Pavilion's tower, which of these two pavilions would be best described as featuring a prominent tower rather than a very steep roof?
   *Image Evidence ID:* 30021436
   *Answer:* The Morocco Pavilion.

4. **(Text)** According to the provided text, what significant historical fact is associated with the Morocco Pavilion regarding its status as a World Showcase addition?
   *Text Evidence ID:*
   d5bef66e0dba11ecb1e81171463288e9_7
   *Answer:* It was the first expansion pavilion to be added to World Showcase, opening on September 7, 1984.

# D BASELINE RAG PERFORMANCE

## D.1 SINGLE-STEP RAG BASELINE

Table 5 reports the results of the fixed-step RAG baseline under the **single-step retrieval** setting. For all experiments, we use the `BAAI/bge-large-en-v1.5` (Zhou et al., 2024) embedding model for dense retrieval. If the input question contains images, the system performs a one-step *image retrieval* to obtain the corresponding caption, which is then concatenated with the query before being fed into the model. If the input is purely textual, the query itself is used for one-step *text retrieval*. In both cases, only the *top-1 retrieved passage* is selected as context and appended to the input prompt.

To ensure a fair comparison with agentic multimodal RAG, we enforce the same evaluation policy: models are only allowed to output an answer when they are sufficiently confident, otherwise they must abstain. This constraint avoids artificially inflating accuracy by forcing guesses and reflects the practical reliability of models in knowledge-grounded settings.

Our results reveal a clear contrast between model families. More robust agentic models (e.g., proprietary Gemini-2.5-Pro) tend to be conservative: when provided with only a single retrieved knowledge piece, they often judge the evidence as insufficient and refrain from answering. In contrast, most open-source models (e.g., InternVL-3.5-Pro, Qwen2.5-VL-7B) display overconfidence, attempting to generate complete answers even when the retrieved context is too limited. While this behavior leads to superficially higher accuracy under fixed-step evaluation, it also exposes a weakness of the fixed RAG paradigm: relying on a single evidence piece frequently fails to support complex reasoning, producing shallow or unsupported answers.

Table 5: Evaluation of various models on the MC-Search benchmark under **single-step RAG**.

| Reason Graphs | Model | Answer Accuracy | | | Chain Alignment | | Golden F1 (↑) |
|---|---|---|---|---|---|---|---|
| | | F1 (↑) | ΔF1 (↑) | LJ (↑) | HPS (↑) | RD (↓) | |
| Text-Only Chain | GPT-4o-Mini | 27.50 | **27.48** | 1.69 | | | 61.90 |
| | Gemini-2.5-Flash | 10.88 | 10.81 | 0.75 | | | **67.72** |
| | Gemini-2.5-Pro | 11.11 | 11.10 | 0.70 | 19.22 | 3.00 | 62.42 |
| | Claude-3.7-Sonnet | 22.71 | 22.09 | 1.69 | | | 66.38 |
| | InternVL3.5-8B | 30.98 | 24.00 | 1.47 | | | 65.72 |
| | Qwen2.5-VL-7B | **35.26** | 26.61 | **1.81** | | | 61.57 |
| Image-Initiated Chain | GPT-4o-Mini | 29.37 | **27.06** | 0.83 | | | 68.29 |
| | Gemini-2.5-Flash | 3.63 | -3.09 | 0.22 | | | 72.39 |
| | Gemini-2.5-Pro | 2.34 | -2.51 | 0.13 | 21.52 | 3.00 | 69.83 |
| | Claude-3.7-Sonnet | 8.29 | 3.58 | 1.18 | | | **72.62** |
| | InternVL3.5-8B | 21.10 | 11.48 | 0.77 | | | 63.86 |
| | Qwen2.5-VL-7B | **40.31** | 22.66 | **1.62** | | | 60.95 |
| Multi-Images Fork | GPT-4o-Mini | 25.53 | 20.69 | 1.27 | | | 59.46 |
| | Gemini-2.5-Flash | 6.49 | 1.58 | 0.56 | | | **64.40** |
| | Gemini-2.5-Pro | 7.26 | 3.47 | 0.51 | 15.38 | 3.00 | 61.29 |
| | Claude-3.7-Sonnet | 12.70 | 10.77 | 1.36 | | | 62.41 |
| | InternVL3.5-8B | 33.11 | **21.01** | 1.59 | | | 57.65 |
| | Qwen2.5-VL-7B | **36.96** | 18.36 | **1.89** | | | 57.04 |
| Parallel Image-Text Fork | GPT-4o-Mini | 15.20 | 14.87 | 0.77 | | | 53.98 |
| | Gemini-2.5-Flash | 4.09 | 2.11 | 0.30 | | | 57.99 |
| | Gemini-2.5-Pro | 4.04 | 3.40 | 0.32 | 15.06 | 3.00 | 53.46 |
| | Claude-3.7-Sonnet | 9.61 | 7.83 | 1.22 | | | 58.51 |
| | InternVL3.5-8B | 20.63 | 13.94 | 0.99 | | | **58.82** |
| | Qwen2.5-VL-7B | **29.70** | 21.08 | **1.57** | | | 55.76 |
| Text-Initiated Chain | GPT-4o-Mini | 24.67 | 13.02 | 1.43 | | | 49.47 |
| | Gemini-2.5-Flash | 16.78 | -0.43 | 1.30 | | | **66.27** |
| | Gemini-2.5-Pro | 14.51 | -0.90 | 1.22 | 26.86 | 3.00 | 55.94 |
| | Claude-3.7-Sonnet | 17.63 | -1.11 | 1.80 | | | 57.13 |
| | InternVL3.5-8B | 28.82 | 15.96 | 1.38 | | | 38.70 |
| | Qwen2.5-VL-7B | **42.97** | 25.84 | **2.48** | | | 39.86 |

## D.2 TRADITIONAL MULTI-MODAL RAG BASELINE

To provide a stronger fixed-step baseline, we extend the single-step RAG setup to a two-hop variant. For multimodal queries, we follow the standard multimodal RAG pipeline (Lin et al., 2023; Joshi et al., 2024; Lin & Byrne, 2022; Zheng et al., 2025a): first, we perform *image retrieval* to obtain the caption of the input image. The retrieved caption is concatenated with the original query, and this combined text is then used to conduct a second-step *text retrieval*. For text-only queries, we directly perform two successive rounds of text retrieval. In both cases, the final retrieved passage (top-1) is appended as context to the input of the multimodal LLM.

As with the single-step baseline, we require models to output an answer only when sufficiently confident, ensuring fair comparison with agentic multimodal RAG and avoiding inflated accuracy due to forced guesses. Table 6 presents the results of the fixed two-hop RAG baseline. Compared to single-step retrieval (Table 5), two-hop retrieval yields consistent improvements across reasoning structures, especially for multimodal cases such as Multi-Images Fork and Text-Initiated chains. This confirms that additional retrieval steps can partially mitigate the information bottleneck of single-hop retrieval by providing more context.

However, performance still lags behind agentic multimodal RAG. While fixed two-hop retrieval provides more evidence, it cannot flexibly adapt the number of retrievals or strategically integrate cross-modal knowledge. In contrast, agentic approaches selectively plan multiple hops and decide when to stop, yielding more faithful reasoning chains and substantially higher overall accuracy. These results highlight both the utility and the limitations of fixed two-hop RAG: it is stronger than single-step retrieval but remains rigid and less reliable than adaptive agentic multimodal RAG.

Table 6: Evaluation of various models on the **MC-Search** benchmark under **traditional MM-RAG**.

| Reason Graphs | Model | Answer Accuracy | | | Chain Alignment | | Golden F1 (↑) |
|---|---|---|---|---|---|---|---|
| | | F1 (↑) | ΔF1 (↑) | LLJ (↑) | HPS (↑) | RD (↓) | |
| Image Initiated Chain | GPT-4o-Mini | 35.61 | **33.30** | 1.58 | 21.11 | 2.42 | 68.29 |
| | Gemini-2.5-Flash | 7.42 | 0.70 | 0.32 | | | 72.39 |
| | Gemini-2.5-Pro | 6.06 | 1.21 | 0.25 | | | 69.83 |
| | Claude-3.7-Sonnet | 21.72 | 21.10 | 1.72 | | | **72.62** |
| | InternVL3.5-8B | 31.26 | 21.64 | 1.24 | | | 63.86 |
| | Qwen2.5-VL-7B | **39.59** | 21.94 | **1.79** | | | 60.95 |
| Multi-Images Fork | GPT-4o-Mini | 27.88 | **23.04** | 1.04 | 19.35 | 2.76 | 59.46 |
| | Gemini-2.5-Flash | 10.29 | 5.38 | 0.56 | | | **64.40** |
| | Gemini-2.5-Pro | 11.03 | 7.24 | 0.57 | | | 61.29 |
| | Claude-3.7-Sonnet | 19.96 | 15.25 | 1.67 | | | 62.41 |
| | InternVL3.5-8B | 34.55 | 22.45 | 1.58 | | | 57.65 |
| | Qwen2.5-VL-7B | **36.61** | 18.01 | **1.97** | | | 57.04 |
| Parallel Image-Text Fork | GPT-4o-Mini | 19.68 | 19.35 | 1.53 | 10.87 | 3.18 | 53.98 |
| | Gemini-2.5-Flash | 5.19 | 3.21 | 0.26 | | | 57.99 |
| | Gemini-2.5-Pro | 4.03 | 3.39 | 0.21 | | | 53.46 |
| | Claude-3.7-Sonnet | 15.80 | 13.87 | 1.45 | | | 58.51 |
| | InternVL3.5-8B | 23.83 | 17.14 | 1.14 | | | **58.82** |
| | Qwen2.5-VL-7B | **30.16** | **21.54** | **1.61** | | | 55.76 |
| Text-Initiated Chain | GPT-4o-Mini | 28.23 | 16.58 | 1.53 | 26.72 | 1.90 | 49.47 |
| | Gemini-2.5-Flash | 20.51 | 3.30 | 1.19 | | | **66.27** |
| | Gemini-2.5-Pro | 17.60 | 2.19 | 1.03 | | | 55.94 |
| | Claude-3.7-Sonnet | 25.55 | 23.77 | 2.20 | | | 57.13 |
| | InternVL3.5-8B | 30.80 | 17.94 | 1.47 | | | 38.70 |
| | Qwen2.5-VL-7B | **41.80** | **24.67** | **2.47** | | | 39.86 |

# E    DATASET STATISTICS

Table 7 provides a detailed breakdown of the MC-Search benchmark. The dataset contains a total of 3,333 multimodal reasoning samples spanning five representative reasoning structures, with the largest portion being Image-Initiated Chains (1,306 samples), followed by Text Chains (945 samples) and Parallel Visual-Textual Forks (680 samples). Although smaller in size, Text-Initiated Chains (169 samples) and Multi-Images Forks (233 samples) introduce challenging cases that require explicit cross-modal or multi-image integration.

The benchmark is grounded in a large-scale knowledge base containing nearly 390K images and 780K textual documents, ensuring broad coverage across both modalities. On average, questions are 33 tokens long, while answers are longer and entity-rich (47 tokens with 8 entities on average), reflecting the complexity of the tasks.

In terms of reasoning depth, the average number of hops varies by structure, ranging from 3.17 in Text-Initiated Chains to around 4 in Parallel Visual-Textual Forks, demonstrating the diversity of reasoning complexity. Across all samples, the majority of hops rely on text retrieval (79.7%), while image retrieval accounts for roughly one-fifth (20.2%), highlighting a balanced yet text-dominant modality distribution.

Table 7: Statistics of MC-Search benchmark.

| Statistic | Number |
| --- | --- |
| Parallel Visual-Textual Fork Samples | 680 |
| Image-Initiated Chain Samples | 1,306 |
| Text Chain Samples | 945 |
| Text-Initiated Chain Samples | 169 |
| Multi-Images Fork Samples | 233 |
| Total Samples | 3,333 |
| Total Images in the Knowledge Base | 389,750 |
| Total Documents in the Knowledge Base | 784,473 |
| Avg. Question Length | 33.23 |
| Avg. Answer Length | 47.40 |
| Avg. Answer Entities | 8.03 |
| Avg. Hops in Parallel Visual-Textual Fork | 4.01 |
| Avg. Hops in Image-Initiated Chain | 3.82 |
| Avg. Hops in Text Chain | 3.65 |
| Avg. Hops in Text-Initiated Chain | 3.17 |
| Avg. Hops in Multi-Images Fork | 3.97 |
| Hops with Text Retrieval | 10,063 (79.72%) |
| Hops with Image Retrieval | 2,550 (20.20%) |

# F    DATASET QUALITY VERIFICATION

To assess dataset quality, we employed **Gemini-2.5-Pro** to score each gold reasoning chain along four dimensions: (i) factual correctness, (ii) step necessity, (iii) clarity, and (iv) multimodal alignment. Scores are integers from 1 (poor) to 5 (perfect), with the overall score defined as the average of the four. Table 8 summarizes the results.

Table 8: Average Gemini-2.5-Pro evaluation scores.

| Dimension | Average Score |
| --- | --- |
| Factual correctness | 4.93 |
| Step necessity | 4.70 |
| Clarity | 4.88 |
| Multimodal alignment | 4.95 |
| Overall (mean) | 4.87 |

## G CROSS-MODEL AND HUMAN CONSISTENCY ANALYSIS

We report an additional consistency study to assess the stability of the LLM-as-Judge (LLJ) evaluation. Agreement across heterogeneous model families and human experts shows that LLJ scoring captures objective and broadly applicable quality criteria, with no observable dependence on the preferred reasoning patterns of the primary judging model (Gemini-2.5-Pro) used in our experiments.

### G.1 CROSS-MODEL LLM-AS-JUDGE CONSISTENCY

We randomly sample five non-overlapping subsets (each 10%) of predictions generated by the Gemini-2.5-Pro backbone in Table 3. Each subset is independently rescored by three **heterogeneous model families**: Gemini-2.5-Pro, GPT-5, and Qwen-VL-Plus. For each pair of judges, we compute Pearson correlation, Spearman rank correlation, Bland–Altman mean bias, and limits of agreement (LoA). Table 9 reports the mean and variance of the consistency score across five subsets.

Table 9: Cross-model LLJ consistency scores, with mean (variance) reported. High correlations and small mean biases indicate strong agreement across evaluator families.

| Evaluator Pair | Pearson | Spearman | Mean Bias | LoA |
|---|---|---|---|---|
| Gemini / GPT-5 | **0.9330** (0.00076) | **0.8720** (0.00120) | +0.0288 (0.00051) | 0.8438 (0.07937) |
| Gemini / Qwen | **0.9147** (0.00116) | **0.8145** (0.00360) | −0.0083 (0.00082) | 0.9295 (0.08834) |
| GPT-5 / Qwen | **0.9147** (0.00131) | **0.8259** (0.00374) | −0.0371 (0.00212) | 0.9341 (0.09565) |

All evaluator pairs show Pearson correlations above 0.91 and Spearman correlations above 0.81. Mean biases remain small in magnitude ($< 0.04$), and LoA ranges are narrow and stable. These results indicate that the LLJ evaluator generates consistent assessments across model families with diverse training data and architectures. Notably, the scores do not display any preferential tendency toward answers produced by Gemini-2.5-Pro, despite it being the primary judge in our main evaluation.

To further illustrate this, we provide Hexbin correlation plots between Gemini-2.5-Pro and GPT-5 across all scoring dimensions (Accuracy, Alignment, Coverage, Coherence, and Entities). The distributions concentrate tightly around the diagonal, confirming that the LLJ scores follow **objective**, **generalizable evaluation criteria** rather than model-specific patterns.

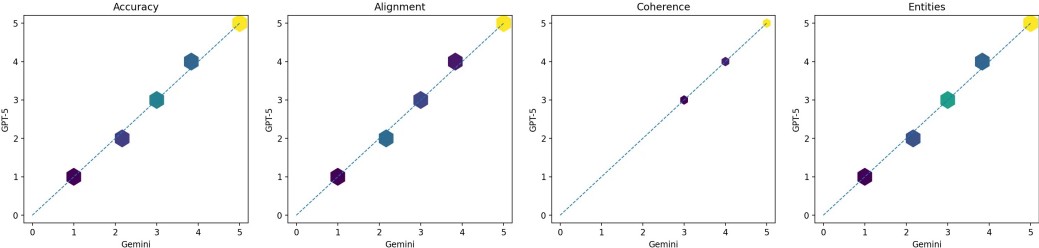

Figure 6: Hexbin plots of Gemini-2.5-Pro vs. GPT-5 scores across four LLJ dimensions.

### G.2 HUMAN AND LLM-AS-JUDGE AGREEMENT

We additionally compare LLJ scores against human annotations collected from computer-science PhD experts on 100 randomly sampled predictions. Humans score each sample along the four LLJ dimensions. Table 10 reports correlations and Bland–Altman statistics between human scores and Gemini-2.5-Pro scores for predictions generated by the Gemini-2.5-Pro backbone.

Overall, the LLJ scores closely track human judgments across all content-grounded dimensions, with Pearson correlations above 0.92 for Accuracy, Entity Coverage, and the Stacked score. Even for the more subjective Coherence dimension, rank-order agreement remains strong (Spearman = 0.82), confirming that the LLJ evaluation provides reliable and human-aligned quality assessments.

Table 10: Human–LLJ agreement scores.

| Dimension | Pearson | Spearman | Mean Bias | LoA |
|---|---|---|---|---|
| Accuracy | **0.9632** | **0.9556** | +0.0800 | 0.6606 |
| Entity Coverage | **0.9375** | **0.8691** | +0.1200 | 0.9309 |
| Coherence | 0.6061 | **0.8200** | −0.0200 | 0.6778 |
| Alignment | **0.9275** | 0.7925 | +0.2750 | 0.9982 |
| Stacked Score | **0.9323** | **0.8675** | +0.1138 | 0.8562 |

### G.3 Cross-Model Ground-Truth Consistency

To examine whether the agentic MM-RAG pipeline depends on model-specific phrasing or syntactic patterns in the reference reasoning chains, we construct an alternative benchmark using **ground truth generated by a different model family**. For each of the five graph topologies, we randomly sample 50 instances and use GPT-5 to regenerate all intermediate sub-answers and the final answer following the construction procedure in Section 2. These GPT-5 chains serve as an additional set of ground-truth annotations.

We then evaluate Gemini-2.5-Pro and GPT-4o-mini under the full agentic MM-RAG pipeline. As shown in Table 11, both models achieve highly **consistent performance** across GPT-generated and Gemini-generated ground truths. F1 and LLJ scores closely track each other across all topologies, and the performance ordering between models remains stable.

These results indicate that the pipeline is robust to variations in the source model used to generate reference reasoning chains. The evaluation reflects genuine multi-hop reasoning ability rather than dependence on stylistic or syntactic artifacts that may favor a specific source model family.

Table 11: Performance of Gemini-2.5-Pro and GPT-4o-mini when evaluated against ground-truth reasoning chains generated by GPT-5 (GPT columns) or by the original Gemini-based construction (Gemini columns).

| Graph Type | Model | F1 Score (↑) | | LLJ Score (↑) | |
|---|---|---|---|---|---|
| | | GPT | Gemini | GPT | Gemini |
| Text Chain | Gemini-Pro | 32.52 | 34.42 | 3.44 | 3.46 |
| | GPT-4o-mini | 26.05 | 26.98 | 3.01 | 3.05 |
| Image-Initiated Chain | Gemini-Pro | 41.37 | 46.57 | 3.82 | 3.81 |
| | GPT-4o-mini | 35.01 | 38.96 | 3.24 | 3.20 |
| Multi-Images Fork | Gemini-Pro | 35.49 | 38.04 | 3.48 | 3.66 |
| | GPT-4o-mini | 17.97 | 18.77 | 2.69 | 2.80 |
| Parallel Image-Text Fork | Gemini-Pro | 36.59 | 38.44 | 3.64 | 3.67 |
| | GPT-4o-mini | 21.09 | 21.97 | 2.58 | 2.54 |
| Text-Initiated Chain | Gemini-Pro | 34.91 | 43.71 | 3.42 | 3.82 |
| | GPT-4o-mini | 22.95 | 26.01 | 2.95 | 3.03 |

## H Fine-Grained Improvement Analysis of SEARCH-ALIGN

To examine how SEARCH-ALIGN alters the process behavior of multimodal agents, we analyze InternVL-3.5-8B before and after alignment using a fine-grained error taxonomy covering planning, retrieval, and modality selection behaviors. For each sample, an external LLM annotates four binary error types: (i) *Retrieval-Failure*, (ii) *Step-Omission*, (iii) *Order/Dependency-Error*, and (iv) *Modality-Mismatch*. We report the proportion of examples exhibiting each error across the MC-SEARCH validation split.

Figure 7 summarizes the results. SEARCH-ALIGN leads to consistent reductions across all four error categories, with the most substantial improvements observed in modality selection and multi-hop

completeness. Modality-Mismatch decreases from 65.75% to 34.72%, indicating that the model learns to choose visual versus textual evidence more appropriately. Step-Omission is also notably reduced (86.15% to 73.66%), suggesting that the agent follows more complete reasoning chains and is less likely to skip intermediate steps. Retrieval-Failure and Order/Dependency-Error show smaller but stable reductions, reflecting improvements in evidence acquisition and hop structuring.

Overall, this process-level analysis shows that SEARCH-ALIGN does not simply encourage imitation of annotated reasoning chains; rather, it induces measurable improvements across multiple dimensions of planning and multimodal decision-making. These reductions in structured error patterns complement the quantitative gains reported in answer accuracy and alignment metrics.

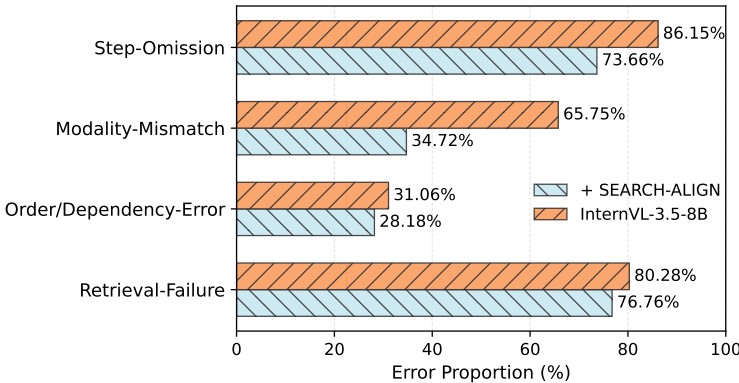

Figure 7: Error proportions before and after SEARCH-ALIGN across four process-level categories.

## I  SEARCH-ALIGN FINE-TUNING DETAILS

To align the reasoning trajectories of multimodal large language models (MLLMs) with golden search chains, we perform supervised fine-tuning (SFT) using the `Llama-factory` framework (Zheng et al., 2024). All experiments are conducted on 4 NVIDIA A100 GPUs (80GB memory each). We report the detailed hyper-parameter configurations for both open-source backbones used in our study, namely InternVL3.5-8B and Qwen/Qwen2.5-VL-7B-Instruct. Table 12 summarizes all fine-tuning hyper-parameters in a unified format.

For clarity, we highlight the following aspects:

1. **Backbone Models.** InternVL uses the official *InternVL3.5-8B* checkpoint, while Qwen adopts the *Qwen/Qwen2.5-VL-7B-Instruct* variant.

2. **Training Setup.** Both models are fine-tuned with identical optimizer and scheduler settings, but differ in learning rate and epoch count due to their different convergence characteristics.

3. **Closed-source Baselines.** For proprietary models (e.g., Gemini-2.5-Pro/Flash, GPT-4.1), no fine-tuning is applied. Their internal *thinking budgets* remain at the default values.

Table 12: Fine-tuning hyper-parameters for Search-Align alignment on InternVL3.5-8B and Qwen/Qwen2.5-VL-7B-Instruct. Both models are trained on 4×A100 GPUs.

| Hyper-parameter | InternVL3.5-8B | Qwen2.5-VL-7B-Instruct |
|---|---|---|
| Optimizer | AdamW | AdamW |
| Learning Rate | 1e-5 | 1e-4 |
| Training Epochs | 1 | 2 |
| Warm-up Ratio | 0.1 | 0.1 |
| Scheduler Type | Cosine | Cosine |
| Train Batch (per device) | 1 | 1 |
| Gradient Accumulation Steps | 8 | 8 |

## J  SOFT HPS EVALUATION WITH THRESHOLDED SEMANTIC MATCHING

The Hit per Step (HPS) metric in the main paper evaluates whether the retrieved evidence exactly matches the gold evidence at each reasoning hop. Because our maximum-weight matching formulation naturally supports similarity-based matching, we further examine a soft HPS variant in which an evidence pair is considered a hit if its embedding similarity exceeds a threshold $\tau$. We report results for four thresholds, $\tau \in \{1.0, 0.95, 0.90, 0.85\}$, corresponding to exact match and increasingly relaxed semantic similarity.

Table 13 shows that soft-HPS scores for both Gemini-2.5-Pro and Gemini-Flash increase as $\tau$ decreases, indicating that models often retrieve semantically related evidence even when exact matches are missed. The improvements become smaller near $\tau = 0.85$, suggesting that the remaining errors are driven by suboptimal planning, such as imperfect question decomposition or missing reasoning steps, which prevents the retrieval of the truly relevant evidence.

Table 13: Soft HPS scores for Gemini-2.5-Pro and Gemini-Flash under different similarity thresholds $\tau$.

| Graph Type | Gemini-2.5-Pro Soft-HPS | | | | Gemini-Flash Soft-HPS | | | |
|---|---|---|---|---|---|---|---|---|
| | $\tau$=1.0 | $\tau$=0.95 | $\tau$=0.90 | $\tau$=0.85 | $\tau$=1.0 | $\tau$=0.95 | $\tau$=0.90 | $\tau$=0.85 |
| Text Chain | 21.59 | 35.59 | 38.05 | 41.26 | 13.33 | 26.86 | 28.78 | 30.97 |
| Image-Initiated Chain | 25.90 | 38.74 | 40.48 | 42.87 | 25.03 | 43.19 | 45.04 | 47.42 |
| Multi-Images Fork | 18.68 | 25.58 | 26.49 | 28.18 | 18.53 | 24.51 | 25.35 | 27.20 |
| Parallel Image-Text Fork | 16.74 | 26.25 | 27.68 | 30.22 | 20.73 | 25.68 | 27.12 | 29.69 |
| Text-Initiated Chain | 19.51 | 26.31 | 27.81 | 29.88 | 21.67 | 29.94 | 30.55 | 32.03 |

## K  EVALUATION UNDER TOP-$k$ RETRIEVAL

This section evaluates whether the retrieval dynamics in the main analysis hold when each step retrieves multiple candidates. Across top-$k$ retrieval settings ($k \in 1, 3, 5$), the overall conclusions on under- and over-retrieval remain unchanged. We also evaluate SEARCH-ALIGN under the same settings and find that its improvements are stable, indicating that both empirical insights generalize reliably to broader retrieval regimes.

### K.1  UNDER- AND OVER-RETRIEVAL UNDER RETRIEVAL@K

We extend the over- and under-retrieval analysis to top-$k$ retrieval by allowing Gemini-2.5-Flash to retrieve the top-$k$ evidence candidates at each step ($k \in \{1, 3, 5\}$). We evaluate $\Delta$F1 across different $\Delta$Step values, where negative values indicate under-retrieval and positive values indicate over-retrieval. Figure 8 reports the results.

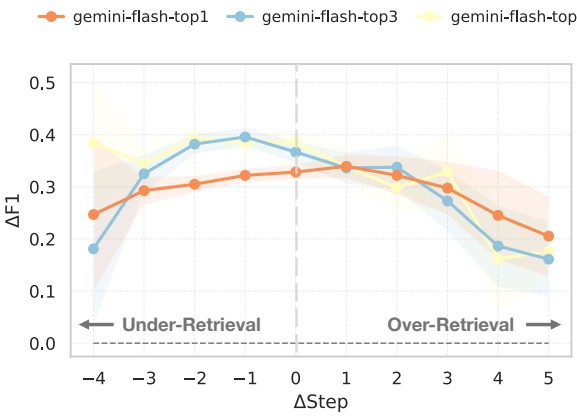

Figure 8: $\Delta$F1 across $\Delta$Step values for Gemini-2.5-Flash under top-$k$ retrieval ($k = 1, 3, 5$).

Across all three settings, the qualitative trend is consistent. Severe under-retrieval ($\Delta$Step $< -3$) causes large performance drops because essential evidence is skipped. As $k$ increases, the peak performance region shifts slightly toward $\Delta$Step near $-1$ or $-2$, suggesting that additional retrieved items help cover relevant evidence even with fewer reasoning steps. Larger $k$ also amplifies the negative effect of over-retrieval: for $\Delta$Step $\geq 4$, performance declines more steeply due to accumulated noisy information. Thus, moderate deviations around the optimal retrieval depth remain beneficial, while strong under- or over-retrieval is consistently detrimental.

### K.2 SEARCH-ALIGN UNDER RETRIEVAL@K

We further evaluate Qwen2.5-VL-7B and its SEARCH-ALIGN variant under the same retrieval settings. Table 14 reports F1, $\Delta$F1, Hit per Step, and Rollout Deviation.

Before alignment, increasing $k$ yields only small F1 improvements, Hit per Step remains low, and Rollout Deviation stays high ($\sim$3–4). This indicates that broader retrieval alone does not resolve planning issues such as missed or misordered reasoning steps.

After applying SEARCH-ALIGN, Qwen2.5-VL-7B shows consistent improvements across all $k$: F1 increases by 15–20 points, Hit per Step rises substantially, and Rollout Deviation decreases to around 1.0. These gains hold for retrieval@1, @3, and @5, and additional candidates provide diminishing but still positive benefits. Overall, SEARCH-ALIGN substantially improves both retrieval accuracy and multi-hop execution, enabling the model to utilize larger candidate sets effectively while maintaining stable reasoning behavior.

Table 14: Performance of Qwen2.5-VL-7B and +SEARCH-ALIGN under Retrieval@1, @3, and @5. We report F1, $\Delta$F1, Hit per Step (HPS), and Rollout Deviation (RD). Best values per topology and metric are highlighted in **bold** font.

| Graph Type | Model | Retrieval@1 | | | | Retrieval@3 | | | | Retrieval@5 | | | |
|---|---|---|---|---|---|---|---|---|---|---|---|---|---|
| | | F1 | $\Delta$F1 | HPS | RD | F1 | $\Delta$F1 | HPS | RD | F1 | $\Delta$F1 | HPS | RD |
| Image-Initiated | Qwen2.5-VL-7B | 26.30 | 8.65 | 16.51 | 4.04 | 29.21 | 11.56 | 16.17 | 3.79 | 29.35 | 11.70 | 16.95 | 3.79 |
| Chain | +SEARCH-ALIGN | **45.70** | **28.05** | **33.59** | **0.70** | **47.19** | **29.54** | **39.45** | **0.75** | **47.29** | **29.64** | **41.79** | **0.79** |
| Multi-Images | Qwen2.5-VL-7B | 26.13 | 7.53 | 12.58 | 3.80 | 30.41 | 11.81 | 12.80 | 3.35 | 30.38 | 11.78 | 14.15 | 3.23 |
| Fork | +SEARCH-ALIGN | **38.01** | **19.41** | **41.20** | **1.16** | **40.00** | **21.40** | **46.82** | **1.21** | **39.80** | **21.20** | **48.21** | **1.28** |
| Parallel | Qwen2.5-VL-7B | 22.94 | 14.32 | 10.09 | 3.93 | 26.28 | 17.66 | 12.62 | 3.39 | 26.41 | 17.79 | 15.24 | 3.26 |
| Image-Text Fork | +SEARCH-ALIGN | **32.73** | **24.11** | **25.07** | **1.05** | **33.54** | **24.92** | **28.43** | **1.09** | **32.77** | **24.15** | **30.93** | **1.18** |
| Text-Initiated | Qwen2.5-VL-7B | 31.47 | 14.34 | 17.02 | 3.80 | 35.54 | 18.41 | 18.86 | 3.20 | 35.25 | 18.12 | 18.83 | 3.14 |
| Chain | +SEARCH-ALIGN | **45.07** | **27.94** | **33.59** | **0.87** | **46.69** | **29.56** | **41.06** | **0.95** | **46.73** | **29.60** | **42.35** | **0.93** |

# L  AGENTIC MM-RAG CASE STUDY

## L.1  SUCCESS CASE

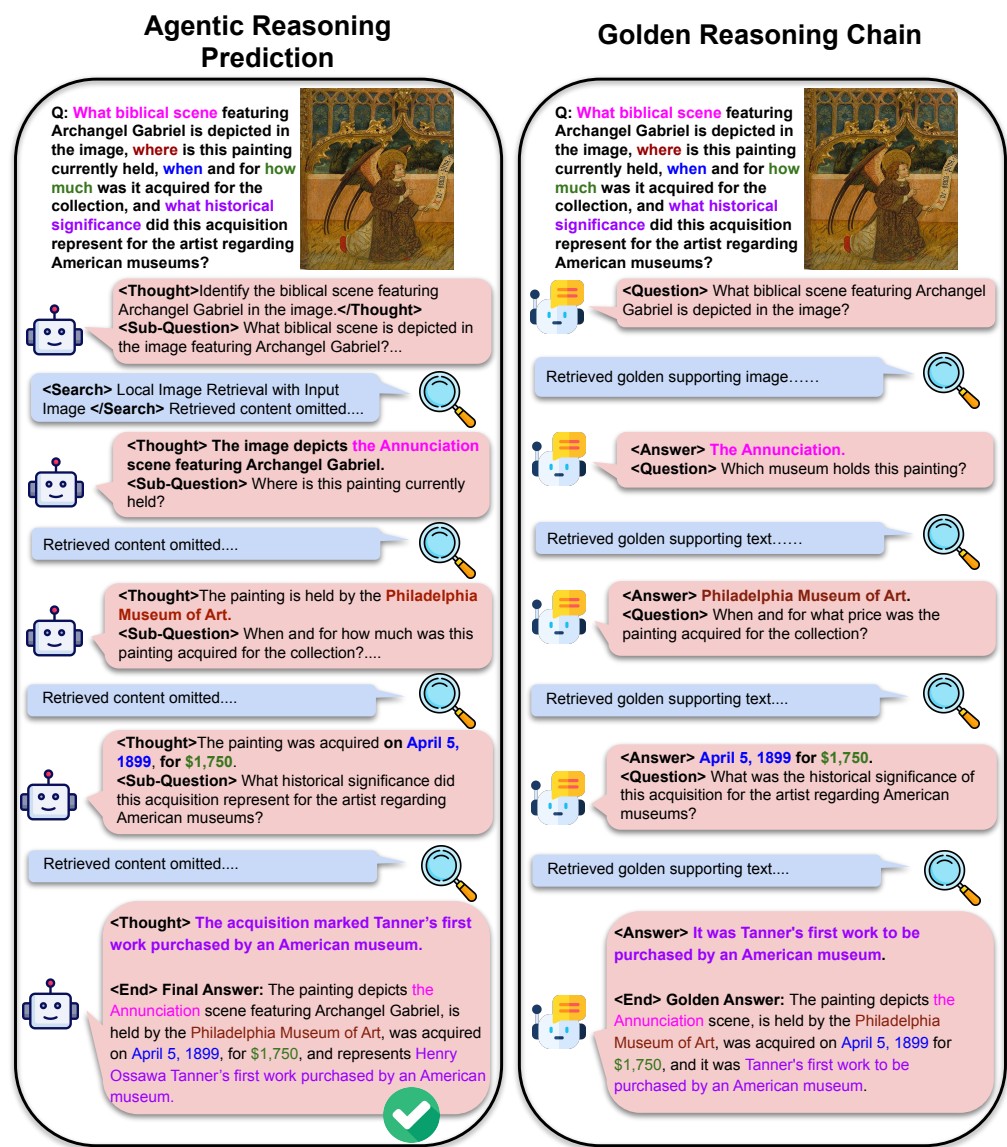

Figure 9: Success case. The reasoning chain produced by the agent aligned with the golden reasoning chain and the final answer includes all key knowledge entities.

## L.2 FAILURE CASE

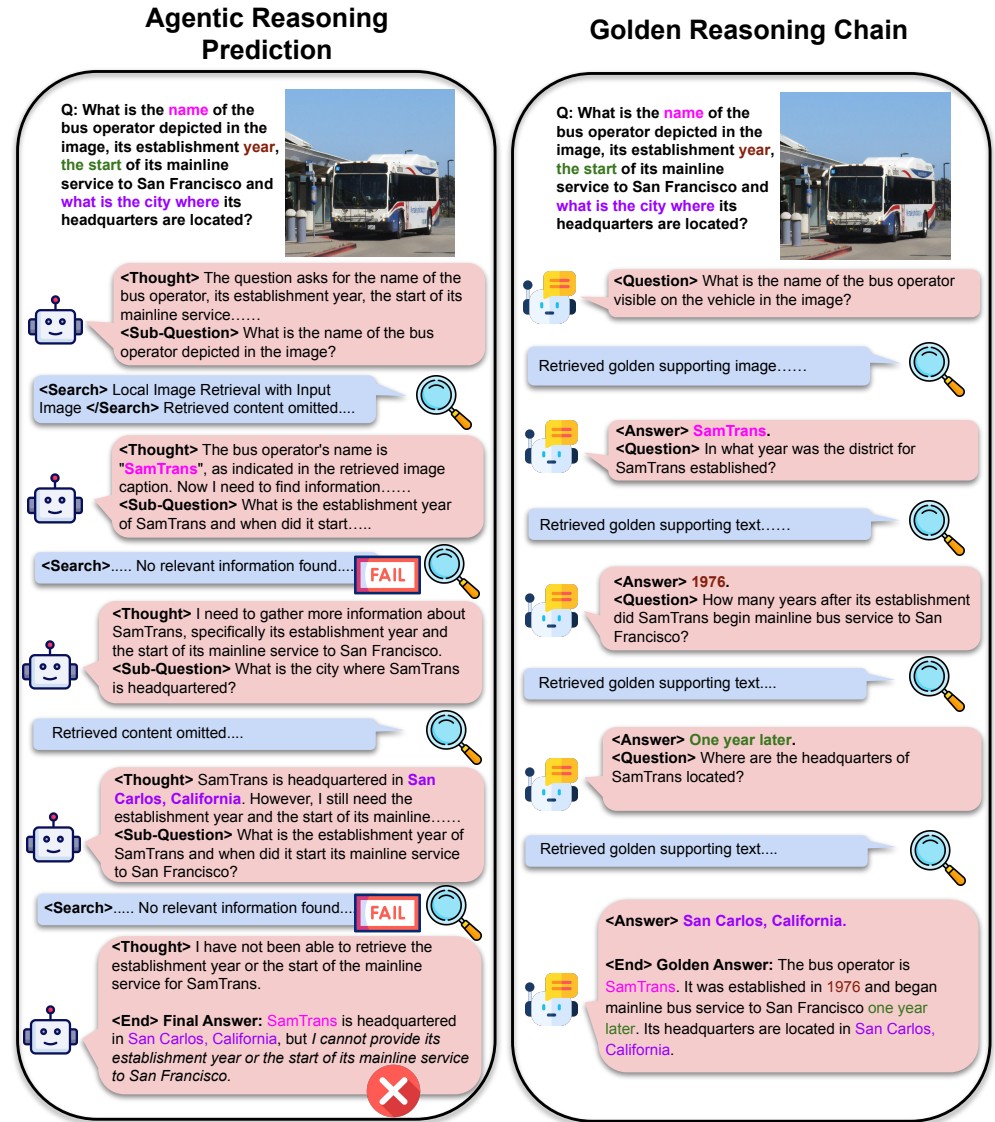

Figure 10: Failure case. While the agent successfully retrieves the first and the last hop information, it fails to retrieve the second and third hop information. Thus, the final answer does not include all key knowledge entities.

# M    THEORETICAL JUSTIFICATION OF TOPOLOGY DESIGNS

In this section, we demonstrate that the five reasoning topologies defined in MC-SEARCH form a **complete canonical basis** for the space of agentic multimodal RAG traces considered in this work.

## M.1    PROBLEM FORMULATION

We model a reasoning trace as a directed acyclic graph $G = (V, E)$ where:

- Each node $v_t$ represents a retrieval step at hop $t$.
- Each node is assigned a modality $m(v_t) \in \{T, I\}$ (Text or Image).
- The structure is constrained to be either **Linear** (a single path) or **Forked** (containing exactly one parallel branching step that merges into an aggregator).

We define the set of valid topologies $\mathcal{T}$ by exhaustively partitioning the space of graphs based on (1) **Initiation Modality** and (2) **Structural Composition**.

## M.2    PROOF BY EXHAUSTION

**Case 1: Linear Structures (Chains)**    Let $G$ be a linear chain $v_1 \to v_2 \to \cdots \to v_n$. The topology is fully determined by the sequence of modalities. We distinguish three mutually exclusive regimes:

1. **Image-Anchored:** The chain begins with visual input ($m(v_1) = I$).
   - *Definition:* **Image-Initiated Chain**. (Covers all traces starting with an image).
2. **Text-Anchored (Unimodal):** The chain begins with text ($m(v_1) = T$) and contains *only* text nodes ($\forall t, m(v_t) = T$).
   - *Definition:* **Text-Only Chain**. (Represents the standard text-RAG baseline).
3. **Text-Anchored (Multimodal):** The chain begins with text ($m(v_1) = T$) but contains at least one subsequent image node ($\exists t > 1, m(v_t) = I$).
   - *Definition:* **Text-Initiated Chain**. (Captures text-to-visual grounding).

**Case 2: Forked Structures (Parallelism)**    Let $G$ contain a parallel branch where nodes $v_a$ and $v_b$ are retrieved independently. The topology is determined by the multiset of branch modalities $S_{branch} = \{m(v_a), m(v_b)\}$.

1. **Heterogeneous Branching:** $S_{branch} = \{I, T\}$.
   - *Definition:* **Parallel Image–Text Fork**. (Captures simultaneous cross-modal evidence integration).
2. **Homogeneous Visual Branching:** $S_{branch} = \{I, I\}$.
   - *Definition:* **Multi-Images Fork**. (Captures comparative visual reasoning).

**Remark on Homogeneous Text Branching** ($S_{branch} = \{T, T\}$)    The final mathematical possibility is parallel text retrieval. Within the scope of **multimodal** agentic search, we treat this case as structurally reducible to the linear *Text-Only Chain*. Since modern LLMs process multiple text documents via context concatenation, a parallel text retrieval step is functionally equivalent to a single multi-document retrieval hop. Therefore, we do not define a distinct topology for text-only branching, preserving the minimality of the set relative to multimodal operations.

## M.3    CONCLUSION

The five definitions above cover all permutations of initiation modality and distinct multimodal branching structures. Thus, the set is **complete** under our assumptions.

# N    Prompt for Data Construction

This appendix provides the system prompt template used to generate multimodal reasoning chains with Google Gemini. The annotator is instructed to construct complex, structurally grounded multi-hop questions following one of five reasoning graph types: Image-Initiated Chain, Parallel Visual-Textual Fork, Text-Initiated Chain, Parallel Multi-Images Fork, and Text Chain. Each generated chain must explicitly build upon previous answers, remain grounded in the provided images and Wikipedia snippets, and output in strict JSON format with unique supporting evidence identifiers.

We provide five reasoning graph templates with guidelines and few-shot examples to standardize multimodal question generation.

## N.1    Image-Initiated Chain

**Structure Description:** Reasoning begins with visual recognition or inference from an image, followed by sequential text-based hops grounded on that visual anchor.

**Guidelines:**

- Begin with a visual observation from the image.

- Use that entity/fact as the anchor for the next step.

- Add 1–2 textual hops depending on the previous answer.

**Example:** Question: Where did the football team, whose players wore red-and-yellow vests during a 2019 match, originally form, and is that location still part of the city they are associated with today? Answer: The team was originally formed in Partick, which is now part of Glasgow. Sub-questions and answers:

- (Image) Which team wore red-and-yellow vests during a 2019 soccer match? → Partick Thistle F.C.

- (Text) Where was Partick Thistle originally formed? → Partick

- (Text) Is Partick still independent from Glasgow? → No, it is now part of Glasgow.

## N.2    Parallel Image-Text Fork

**Structure Description:** Reasoning starts with a sub-question that requires jointly leveraging both visual and textual information. These signals must be fused before continuing via text.

**Guidelines:**

- Design a sub-question requiring combination of both image and text.

- The fused result forms the anchor for continued reasoning.

- Add a follow-up text hop grounded in the first answer.

**Example:** Question: Is the curved architecture of the Petersen Events Center well-suited for its use as a modern basketball arena at Pitt? Answer: Yes, the curved structure complements its role as a large multi-purpose sports venue. Sub-questions and answers:

- (Image) Does the Petersen Events Center have a curved architectural design suitable for large gatherings? → Yes, the central structure is rounded and accommodates large interior space.

- (Text) What is the primary use of the Petersen Events Center on the Pitt campus? → It is a multi-purpose arena used by the Pitt basketball team.

- (Text) Does the architectural design align with its use as a modern sports arena? → Yes, its rounded shape supports audience visibility and modern sporting needs.

### N.3 TEXT-INITIATED CHAIN

**Structure Description:** Reasoning is initiated via text and ends with image confirmation; the final hop depends on visual content.

**Guidelines:**

- Start with a sub-question answerable via text.
- Use that answer as the anchor for another textual hop.
- The final step should rely on visual confirmation from the image.

**Example:** Question: Is the area in front of the Torre del Reloj enhanced with any water-related architectural features? Answer: Yes, there is a fountain placed in front of the Torre del Reloj, adding a water-related feature to the plaza. Sub-questions and answers:

- (Text) Where is the Torre del Reloj and what landmark is it? → It is a historic clock tower in Chiclana's Plaza Mayor.
- (Image) Is there any water-related structure in front of the Torre del Reloj? → Yes, a fountain is clearly visible.

### N.4 MULTI-IMAGES FORK

**Structure Description:** Reasoning begins by comparing or aggregating information from two images, then follows with text-based interpretation.

**Guidelines:**

- Compare or verify a shared visual pattern across two related images.
- Use their joint interpretation as the basis for a follow-up textual inference.

**Example:** Question: What visual marker appears along both Salou Boulevard and the adjacent plaza, and what does it symbolize? Answer: Blue footprints appear in both places, symbolizing a historical walking route. Sub-questions and answers:

- (Image) What pattern is painted down the center of Salou Boulevard? → Blue footprints.
- (Image) Does the plaza image show the same pattern? → Yes, the same blue footprints.
- (Text) According to local guides, what do blue footprints represent? → They mark a historic promenade route.

### N.5 TEXT CHAIN

**Structure Description:** A standard multi-hop reasoning chain entirely over textual content.

**Guidelines:**

- Construct a multi-hop reasoning chain using only text.
- Each sub-question must depend explicitly on the previous answer.
- Avoid visual modality entirely.

**Example:** Question: Which historical street in Quebec City was known for having a unique wooden paving method in the early 20th century? Answer: Little Champlain Street was known for its board paving in 1916. Sub-questions and answers:

- (Text) What notable historic streets exist in Old Quebec? → Saint-Vallier Est Street and Petit-Champlain (Little Champlain).
- (Text) Which of them had unusual wooden paving in 1916? → Little Champlain Street had board paving.
- (Text) Was board paving common or unique for that era in Quebec City? → It was unique/uncommon, making the street notable.

**Example: Generating Multimodal Reasoning Chains with Gemini**

**Input Prompt:**
You are a multimodal data annotation expert. You are given a multihop multimodal QA generation task.
You are provided with:

- 1 or 2 images (embedded as Base64 above with captions below)
- Several Wikipedia passages
- An original 1-hop simple QA pair for reference

Your task is to generate a **new**, complex, coherent, and structurally grounded multi-hop question and reasoning chain that follows the specified reasoning graph type.

**Reasoning Graph Information:**

- **Graph Type**: {graph_type}
- **Structure Description**: {graph_structure_description}
- **Structure Instruction**: {graph_structure_instruction}
- **Expected Reasoning Hop Count**: 4–5 hops

**Guidelines:**

- The question must align with the given graph structure, integrating visual and textual modalities where required.
- Each sub-question must explicitly build upon the *previous answer*, using it as an anchor for the next hop.
- Each `supporting_fact_id` must be unique across the chain.
- Answers must be directly grounded in the provided images or text snippets (no hallucinated facts).
- The chain must be coherent and progressive, forming a natural reasoning path.
- The final answer should logically follow from the entire chain.
- Each sub-question must specify:
  - `modality`: "text" or "image"
  - `supporting_fact_id`: the ID of the text snippet or image
  - `answer`: the grounded answer
- Output must be in strict JSON format (not Markdown or free text).

**Additional Context:**

- Original question (for inspiration): {entry['Q']}
- Original answer: {entry['A']}
- Image captions: {captions}
- Wikipedia snippets: {text_snippets}

**Expected Output Format:**

```
{
  "question": "...",
  "answer": "...",
  "image_id": ... ,                    # or "image_ids": [...]
  "graph_type": "{graph_type}",
  "subqa_chain": [
    {
      "subquestion": "...",
      "modality": "text" | "image",
      "supporting_fact_id": "...",
      "answer": "..."
    },
    ...
  ]
}
```

## O    PROMPT TEMPLATE FOR AGENTIC MM-RAG PIPELINE

This appendix provides the full system prompt used for agentic MM-RAG pipeline. The assistant is constrained to operate strictly on retrieved local contexts from external text and image knowledge bases, and follows a structured reasoning protocol consisting of `<Thought>`, `<Sub-Question>`, `<Search>`, and `<End>` stages. This ensures that every generated answer is grounded, step-wise, and verifiable.

---

**Gemini Prompt for Multimodal QA**

**System Prompt:**
You are a helpful multimodal QA assistant that works with an external text knowledge base and an external image knowledge base. You must use only the knowledge from the external resources provided. You must **not** retrieve by yourself. All answers must be grounded only in the retrieved local context that I give to you.

**Available retrieval actions:**

- Image Retrieval with Input Image
- Text Retrieval with a specific query
- Image Retrieval with Text Query

**Protocol:**

- `<Thought>` Analyze the original question and decide the next minimal sub-question solvable in one step.
- `<Sub-Question>` Write one sub-question that should be solved in one step (no references).
- `<Search>` Choose exactly one of the retrieval actions above (or "No Retrieval").
- Repeat `<Thought>`/`<Sub-Question>`/`<Search>` zero or more times.
- `<Thought>` Integrate retrieved content and reason to the final answer.
- `<End>` Final Answer: one-sentence answer to the original question.

**Extra Notes:**

1. The user will perform the actual search after you provide the query.
2. The local text knowledge base contains short factual passages. Retrieval returns at most one passage each time.
3. Answers must be grounded **only** in retrieved content.
4. If retrieval is insufficient, continue with additional steps.
5. Keep steps concise; do not include citations or URLs.
6. Do not hallucinate; if not enough information is retrieved, state that you cannot answer.

**Input Format:** `Input Question:   {QUESTION}`

---

## P  PROMPT TEMPLATE FOR LLM-AS-JUDGE

This appendix provides the exact system prompt used to evaluate predicted answers against golden references under an input-only policy. The judge does not assess the truthfulness of the golden answer; it scores the prediction's Accuracy, Entity Coverage, Coherence, and Alignment on 0–5 integer scales following the band definitions. When a prediction claims "insufficient information" while a golden answer exists, it is penalized. The judge outputs strict JSON containing only the four scores, enabling reliable programmatic aggregation for large-scale evaluation.

---

**Example: Judge System Prompt**

**Input Prompt:**
You are a strict, neutral judge. Compare *pred_answer* to *gold_answer* (and optional subchains) for the given question. Use only the provided text; no external knowledge. Do not judge the truth of the gold answer—only the prediction's consistency with it. If the prediction claims "insufficient information" while a gold answer exists, penalize. Keep the output terse.
**Inputs:**

- question
- gold_answer, pred_answer
- gold_subchain, pred_subchain

**Guidelines (Scoring Dimensions 0–5):**

- **Accuracy**: agreement with gold on facts/relations; answers the question; no contradictions.
- **Entity Coverage**: presence and correctness of key entities, values, and relations required by gold.
- **Coherence**: clear stepwise reasoning (if any); no leaps or conflicts; conclusion follows logically.
- **Alignment**: uses only the given information; matches gold subchain when relevant; no unsupported additions.

**Score Bands:** 5 = perfect; 4 = minor issues; 3 = noticeable gaps; 2 = major gaps or weak; 1 = mostly wrong; 0 = off-topic or none.

**Expected Output Format:**

```
{
  "scores": {
    "accuracy": 0,
    "entities": 0,
    "coherence": 0,
    "alignment": 0
  }
}
```

---

