# OpenReview forum: "MC-Search: Evaluating and Enhancing Multimodal Agentic Search with Structured Long Reasoning Chains"
_ICLR.cc/2026/Conference — ICLR 2026 Oral_

### Official Review · Reviewer_mT9g · 2025-10-14

**Soundness:** 3
**Presentation:** 2
**Contribution:** 2
**Rating:** 4
**Confidence:** 4

**Summary:**

This paper introduces MC-SEARCH, a comprehensive benchmark designed to evaluate and advance agentic multimodal retrieval-augmented generation (MM-RAG) systems, particularly focusing on structured, stepwise long-horizon reasoning. MC-SEARCH comprises 3,333 high-quality, HAVE-verified examples encompassing five distinct multi-hop reasoning topologies (including serial, parallel, and cross-modal forks), each annotated with granular sub-questions, modalities, evidence, and intermediate answers. The paper also proposes novel process-level evaluation metrics that go beyond traditional answer accuracy, enabling fine-grained analysis of stepwise retrieval, planning, and reasoning. Extensive benchmarking is conducted on six leading MLLMs, and a process-supervised fine-tuning scheme (SEARCH-ALIGN) leveraging the new dataset is presented, showing notable improvements for open-source models.

**Strengths:**

**Rigorous Dataset Construction:** MC-SEARCH addresses a clear gap in existing multimodal RAG benchmarks by providing long, structured, and verified reasoning chains spanning five rich reasoning topologies. The dataset construction meticulously filters for non-redundancy and necessity at each reasoning step using the HAVE protocol, resulting in high annotation quality

**Process-Level Evaluation:** The paper moves beyond answer-level benchmarking by introducing stepwise “Hit per Step,” Rollout Deviation, and LLM-as-a-Judge metrics, directly quantifying the chain of reasoning, retrieval fidelity, and error types. This is a step change from prior black-box evaluation paradigms.

New Method: They propose SEARCH-ALIGN, a process-supervised fine-tuning framework for MLLMs that leverages verified reasoning chains to align model behavior

**Weaknesses:**

- Although the model coverage is broad, several open-source baselines used are not state-of-the-art for their respective modalities, which may overstate SEARCH-ALIGN’s improvements. The paper does not sufficiently justify the exclusion of other competitive open-source MLLMs or recent RAG architectures such as UniRAG and MIRAGE from direct comparison.

- While Table 3 highlights clear performance gains from SEARCH-ALIGN, the contribution of individual components—planning, retrieval, and modality selection—is analyzed only qualitatively. A more rigorous ablation is needed to determine whether the model genuinely learns better planning strategies or simply imitates supervised reasoning steps.

- Given the use of LLMs for data construction, filtering, and evaluation, to what extent might there be annotation artifacts or overfitting to the verifier’s error modes? How do the authors ensure the robustness of results in light of potential circularity?

**Questions:**

- I’m a bit confused about why the reasoning process is represented as a graph rather than a chain.

- Why didn’t you include ROUGE and MRFS scores [1] in your evaluation?

- Why didn’t you incorporate retrieval-related metrics in your benchmark analysis?

- How does your method Search-ALign perform compared to other RL-based agentic optimization approaches?



[1] Pan Z, Luo H, Li M, et al. Chain-of-action: Faithful and multimodal question answering through large language models[J]. arXiv preprint arXiv:2403.17359, 2024.

---

> ### Author Response · Authors · 2025-11-21
>
> **Thank you for the detailed feedback! We've conducted new experiments to address your concerns and updated the paper accordingly. We provide our response in the form of Q&A.**
>
> ---
>
> ## Q1: Comparing with recent RAG architectures (e.g., UniRAG)
>
> **A1:** MC-SEARCH evaluates **step-adaptive multimodal agents**, whereas UniRAG-style systems use **fixed-step multimodal RAG**. To ensure fairness, we include both a **single-step RAG baseline** and a **strong two-step multimodal RAG baseline** that follows the core UniRAG idea but is upgraded for our setting (image retrieval followed by caption-assisted text retrieval for few-shot examples). A full comparison with both baselines is provided in **[Appendix D](https://openreview.net/pdf?id=JEGDp1E4OH#page=20)**.
>
>
> **Key finding:** Even with two-step multimodal retrieval, fixed RAG pipelines **still underperform agentic multimodal reasoning** across F1, HPS, and RD. In this setting, strong proprietary models such as Gemini-Pro frequently refuse to answer when evidence is incomplete, whereas agentic models continue reasoning and actively retrieve the missing information. When both models are further trained with **SEARCH-ALIGN**, the performance gap becomes larger, showing that process-level supervision benefits **adaptive multi-step agents far more than fixed pipelines in our setting**.
>
>
> **Table: Multimodal RAG baselines (e.g., UniRAG) on different model backbones vs. MMRAG with SEARCH-ALIGN**
>
> | Reason Graph | Model | F1↑ | ΔF1↑ | LLJ↑ | HPS↑ | RD↓ |
> |-----|--------|------|--------|--------|--------|--------|
> | **Image-Initiated Chain** | GPT-4o-Mini | 35.61 | 33.30 | 1.58 | 21.11 | 2.42 |
> |  | Gemini-Pro | 6.06 | 1.21 | 0.25 | 21.11 | 2.42 |
> |  | Claude-Sonnet | 21.72 | 21.10 | 1.72 | 21.11 | 2.42 |
> |  | InternVL3.5-8B | 31.26 | 21.64 | 1.24 | 21.11 | 2.42 |
> |  | **InternVL3.5-8B + SEARCH-ALIGN** | **42.27** | **32.65** | **2.53** | **32.49** | **0.94** |
> |  | Qwen2.5-VL-7B | 39.59 | 21.94 | 1.79 | 21.11 | 2.42 |
> |  | **Qwen2.5-VL-7B + SEARCH-ALIGN** | **45.70** | **28.05** | **2.23** | **33.59** | **0.70** |
> | **Multi-Images Fork** | GPT-4o-Mini | 27.88 | 23.04 | 1.04 | 19.35 | 2.76 |
> |  | Gemini-Pro | 11.03 | 7.24 | 0.57 | 19.35 | 2.76 |
> |  | Claude-Sonnet | 19.96 | 15.25 | 1.67 | 19.35 | 2.76 |
> |  | InternVL3.5-8B | 34.55 | 22.45 | 1.58 | 19.35 | 2.76 |
> |  | **InternVL3.5-8B + SEARCH-ALIGN** | **36.53** | **24.43** | **2.49** | **39.33** | **1.67** |
> |  | Qwen2.5-VL-7B | 36.61 | 18.01 | 1.97 | 19.35 | 2.76 |
> |  | **Qwen2.5-VL-7B + SEARCH-ALIGN** | **38.01** | **19.41** | **2.11** | **38.01** | **1.16** |
> | **Parallel Image-Text Fork** | GPT-4o-Mini | 19.68 | 19.35 | 1.53 | 10.87 | 3.18 |
> |  | Gemini-Pro | 4.03 | 3.39 | 0.21 | 10.87 | 3.18 |
> |  | Claude-Sonnet | 15.80 | 13.87 | 1.45 | 10.87 | 3.18 |
> |  | InternVL3.5-8B | 23.83 | 17.14 | 1.14 | 10.87 | 3.18 |
> |  | **InternVL3.5-8B + SEARCH-ALIGN** | **30.72** | **24.03** | **2.29** | **23.55** | **1.45** |
> |  | Qwen2.5-VL-7B | 30.16 | 21.54 | 1.61 | 10.87 | 3.18 |
> |  | **Qwen2.5-VL-7B + SEARCH-ALIGN** | **32.73** | **24.11** | **1.78** | **25.07** | **1.05** |
> | **Text-Initiated Chain** | GPT-4o-Mini | 28.23 | 16.58 | 1.53 | 26.72 | 1.90 |
> |  | Gemini-Pro | 17.60 | 2.19 | 1.03 | 26.72 | 1.90 |
> |  | Claude-Sonnet | 25.55 | 23.77 | 2.20 | 26.72 | 1.90 |
> |  | InternVL3.5-8B | 30.80 | 17.94 | 1.47 | 26.72 | 1.90 |
> |  | **InternVL3.5-8B + SEARCH-ALIGN** | **40.87** | **28.01** | **2.59** | **37.82** | **1.44** |
> |  | Qwen2.5-VL-7B | 41.80 | 24.67 | 2.47 | 26.72 | 1.90 |
> |  | **Qwen2.5-VL-7B + SEARCH-ALIGN** | **45.07** | **27.94** | **2.42** | **33.59** | **0.87** |
>
>
>
>
> ---
>
> ## Q2: Component-wise effect of SEARCH-ALIGN (better planning or imitation)?
>
> **A2:** In short, SEARCH-ALIGN improves the three core capabilities: planning, retrieval, and modality selection, instead of simply imitating the annotated answers. We conduct a fine-grained process-level error analysis with Gemini-2.5-Pro on InternVL-3.5-8B (see details in [Appendix H](https://openreview.net/pdf?id=JEGDp1E4OH#page=24)), which shows clear reductions across all related error categories:
>
> - **Planning depth and completeness** (lower Step-Omission)
> - **Planning structure and ordering** (lower Order/Dependency-Error)
> - **Retrieval accuracy** (lower Retrieval-Failure)
> - **Correct selection of visual or textual evidence** (lower Modality-Mismatch)
>
> | Error Type               | Before | After (+SEARCH-ALIGN) | Δ ↓ |
> |----------|--------|------|-----|
> | Retrieval-Failure        | 80.28% | 76.76%                 | 3.52 |
> | Step-Omission            | 86.15% | 73.66%                 | 12.49 |
> | Order/Dependency-Error   | 31.06% | 28.18%                 | 2.88 |
> | Modality-Mismatch        | 65.75% | 34.72%                 | 31.03 |
>
> **These targeted reductions show that SEARCH-ALIGN does not simply encourage copying annotated chains. It strengthens planning ability, improves retrieval reliability, and leads to more appropriate use of image and text information.**

---

> ### Author Response · Authors · 2025-11-21
>
> ## Q3: Risk of annotation artifacts or circularity due to LLM involvement?
>
> **A3:** We fully agree that any benchmark built with LLM assistance must be rigorously checked for circularity or construction-model bias. To directly address this risk, we conduct a set of **three complementary robustness studies** that target the exact failure modes the reviewer is concerned about: dependence on the construction model, dependence on the evaluator’s preference, and overfitting to that evaluator’s artifacts.
>
> Across **cross-model judges**, **human–LLM comparisons**, and **ground-truth regeneration**, we find **no evidence** that MC-SEARCH favors Gemini or any construction model’s style. Instead, evaluation signals consistently reflect **model-agnostic, human-aligned reasoning quality**.
>
> Specifically:
> - **Cross-model agreement tests** show that LLM-as-Judge (LLJ) scores remain stable across Gemini-2.5-Pro, GPT-5, and Qwen-VL-Plus.
> - **Human expert studies** confirm LLJ decisions correlate strongly with human judgment.
> - **Ground-truth regeneration using GPT-5** preserves model ranking and performance gaps, ruling out dependence on Gemini’s style.
>
> Together, these results demonstrate that MC-SEARCH’s evaluation pipeline is **not tied to any specific construction model**, and the LLJ-based scoring captures **human-grounded reasoning quality**, not stylistic artifacts.
>
> ### **All supporting evidence and tables appear in our response to **[Reviewer YFCi Q2](https://openreview.net/forum?id=JEGDp1E4OH&noteId=kIeoUe0QJw)**, with full experimental results detailed in [Appendix G](https://openreview.net/pdf?id=JEGDp1E4OH#page=23).**
>
> ---
>
> ## Q4: Why model reasoning process as a graph rather than a chain?
>
> **A4:** The reasoning process is represented as a graph rather than a simple chain because **the step index reflects only the execution order, not the underlying reasoning dependencies**. These dependencies can be serial or parallel, which naturally form **directed acyclic graphs (DAGs)** instead of linear sequences.
>
> **Serial case:** Some topologies are linear, such as Text-Initiated Chains, where each hop depends on the previous one.
>
> **Parallel case:** Other topologies contain parallel branches. For example, in a Multi-Images Fork, the agent may retrieve Image A then Image B, or retrieve Image B then Image A, before comparing them. The retrieval order may vary, but the underlying reasoning structure is the same, forming a branch–merge DAG rather than a chain.
>
> Because these branches are order-permutable, we use maximum-weight bipartite matching to recover the correct structure regardless of execution order.
>
> Thus, execution is sequential, but the reasoning dependencies form true DAGs. [Appendix M](https://openreview.net/pdf?id=JEGDp1E4OH#page=30) further shows that the five topologies constitute a complete canonical basis for multimodal search-enhanced reasoning traces under our assumptions.
>
>
>
> ---
>
> ## Q5: Why didn’t you include ROUGE and MRFS in your evaluation?
>
> **A5:** Thank you for highlighting Chain-of-Action [1]. **We have added it to the Related Work**, particularly its scope in retrieval-augmented multimodal QA. Regarding evaluation, our metric choices follow the design of MC-SEARCH benchmark:
>
> 1. **MC-SEARCH answers are short factual spans rather than long paragraph summaries.**
>    ROUGE-L is designed for paragraph-level summarization, whereas **token-level F1 (as used in Dyn-VQA and similar QA benchmarks)** is more suitable for our short (typically one-sentence) factual answers.
>
> 2. **LLM-as-Judge (LLJ)** provides complementary high-level assessments, including answer accuracy, entity coverage, chain alignment, and coherence, that token-level metrics (e.g., F1, ROUGE, and MRFS) cannot capture.
>
> 3. To directly address your question, we **computed ROUGE-L and MRFS** following [1]. The results are shown in the Table below.
>    **All metrics show highly consistent patterns:** graph types with higher F1 also score higher on ROUGE-L, MRFS, and LLJ, confirming that our main metrics already capture the underlying performance trends.
>
>
> | Graph Type | F1 | LLJ | ROUGE-L | MRFS |
> |-----|-----|------|----|---|
> | Text Chain | 34.03 | 2.49 | 31.50 | 1.41 |
> | Image-Initiated Chain | 44.10 | 3.01 | 43.64 | 1.44 |
> | Multi-Images Fork | 36.80 | 2.35 | 34.28 | 1.41 |
> | Parallel Image-Text Fork | 29.92 | 2.58 | 26.91 | 1.32 |
> | Text-Initiated Chain | 43.55 | 3.30 | 38.20 | 1.42 |
>
> [1] Chain-of-Action: Faithful and Multimodal Question Answering through Large Language Models

---

> ### Author Response · Authors · 2025-11-21
>
> ## Q6: Why didn’t you incorporate retrieval-related metrics in your benchmark analysis?
>
> **A6:** In short, MC-SEARCH already provides **retrieval-focused metrics and analyses by design.** Retrieval is a core part of MC-SEARCH, and retrieval evaluation is already built into our benchmark.
>
> - **Hit per Step (HPS)** provides **topology-aware retrieval evaluation** and is reported in the main results (Table 3).
> - We analyze **retrieval dynamics** in the paper (under-/over-retrieval, optimal retrieval  steps).
> - **Additional retrieval metrics (Retrieval@k and Soft-HPS)** are included in the [Appendix J and K](https://openreview.net/pdf?id=JEGDp1E4OH#page=26), confirming the same conclusions.
>
>
>
>
>
> ---
>
> ## Q7: How does SEARCH-ALIGN compare to RL-based agentic optimization methods?
> **A7:** Our work focuses on **benchmark design, process-level annotations, and evaluation**, rather than proposing a new RL algorithm. SEARCH-ALIGN is included only as a **step-wise supervised SFT baseline** to show how models benefit from MC-SEARCH’s step-wise signals. MC-SEARCH is intentionally designed to **enable and support future RL-based works**:
> - It provides **step-wise sub-questions, evidence, and intermediate answers** that are directly compatible with process-reward modeling in RL.
> - Metrics such as **HPS and RD** offer **fine-grained reward signals** that advanced RL methods (e.g., GRPO) can utilize.
>
> ---
> ## Happy to have further discussion!
> **Thank you again for the thoughtful review. We’ve dedicated many efforts to get the new results and will include them to enhance the paper’s quality. We hope our responses address your concerns, and we are happy to discuss if you have any further questions!**

---

> > ### Comment · Reviewer_mT9g · 2025-11-21
> >
> > I am pretty happy that the author already response all of my concerns. In the first round of review, I raised a serious concern about the benchmark experiment numbers because this paper is a benchmark paper. Now, i think the author provide a huge effort on the rebuttal period to provide a huge experiments and I will support the paper to be accept. As a result, I will raise my score to 6

---

> ### Author Response · Authors · 2025-11-21
>
> Thank you very much for your thoughtful review and encouraging feedback! It means a lot to us :) We are glad that our additional experiments addressed your concerns, and we will incorporate these new results into the paper. Thank you again for your support!
>
> Best wishes,
>
> #13995 Authors

---

### Official Review · Reviewer_YFCi · 2025-10-31

**Soundness:** 2
**Presentation:** 3
**Contribution:** 2
**Rating:** 4
**Confidence:** 3

**Summary:**

This paper introduces MC-SEARCH, a new benchmark for evaluating multimodal agentic Retrieval-Augmented Generation (MM-RAG). The authors argue that existing benchmarks are too simple, focusing on short reasoning tasks. To address this, they make four main contributions:

1. The MC-SEARCH benchmark, a dataset of 3,333 examples featuring long reasoning chains (avg. 3.7 hops) organized into five distinct reasoning topologies.

2. A data filtering process called HAVE (Hop-wise Attribution and Verification of Evidence) to ensure each reasoning step is necessary and non-redundant.

3. New process-level metrics (e.g., Hit per Step, Rollout Deviation) to evaluate intermediate reasoning steps, not just the final answer.

4. The SEARCH-ALIGN framework, a process-supervised fine-tuning method that uses the benchmark's annotations to improve the capabilities of open-source models.

**Strengths:**

The paper's primary strengths lie in its thoughtful benchmark design and its focus on process-level evaluation.

Structured Benchmark Design: The introduction of five explicit reasoning topologies is a significant contribution. It moves beyond simply creating "long" chains and provides a structured way to diagnose specific model failures (e.g., a model failing on Parallel Forks but succeeding on Linear Chains). This offers a more granular analysis than existing benchmarks.

Rigorous Data Curation: The HAVE process is a commendable effort to improve benchmark quality. By programmatically filtering out trivial or redundant reasoning steps, the authors have likely created a more challenging and reliable testbed for genuine reasoning abilities, addressing a common weakness in synthetically generated datasets.

Process-Oriented Evaluation: The push for metrics beyond final answer accuracy is timely. In complex agentic tasks, intermediate failures are often hidden. Metrics like Hit per Step (HPS) and Rollout Deviation (RD) provide a clearer view of how and where models fail, which is critical for future development.

Demonstrated Training Utility: The inclusion of the SEARCH-ALIGN framework is a strong point. By showing that the detailed annotations can be used for process-supervised fine-tuning to improve open-source models, the authors prove that MC-SEARCH is a valuable resource for both evaluation and model development.

**Weaknesses:**

The paper's contributions are undermined by significant weaknesses, primarily an overstatement of novelty and key methodological limitations.

Novelty overstatement — The paper's central claim of being the "first benchmark for agentic MM-RAG with long, step-wise annotated reasoning chains" is not well-supported. Prior work such as Dyn-VQA benchmark was specifically designed for dynamic, multi-hop questions requiring complex, adaptive retrieval and also introduced a self-adaptive planning agent. The paper must reposition its contribution not as being the first, but as providing uniquely structured and verified reasoning chains, and it needs to properly differentiate itself from Dyn-VQA and other related works like WebQA, MRAG-Bench, and MMSearch.

Potential model bias — While the HAVE pipeline partially mitigates bias via cross-model filtering (Qwen2.5-VL and Gemini-Pro), Gemini models still dominate generation, filtering, and evaluation phases. This may tune the benchmark toward Gemini’s reasoning style.

Simplified retrieval setting — The evaluation uses a top-1 retrieval setup, which does not reflect realistic RAG conditions where irrelevant documents co-exist. The conclusions regarding over/under-retrieval and SEARCH-ALIGN efficacy may not fully generalize to top-k retrieval.

Limited justification for reasoning topologies — Although the five topologies are intuitive, the paper does not provide empirical or theoretical justification for their selection (e.g., frequency in real-world tasks). Including such evidence would improve the framework’s validity.

Metric rigidity — The Hit per Step (HPS) metric relies on exact evidence matching, which could penalize models finding semantically equivalent alternatives. The authors do mention a semantic alignment mechanism for structural comparison, but an integrated soft-matching variant could make evaluation fairer.

**Questions:**

How does MC-SEARCH differ empirically and conceptually from Dyn-VQA and MMSearch beyond having predefined topologies? Could the authors quantify the added diagnostic value of these structures?

How do the authors ensure benchmark neutrality given that Gemini models are used for both generation and evaluation? Have they tested whether non-Gemini models (e.g., GPT-4o, Claude, QwenVL) are unfairly disadvantaged?

How might the conclusions about SEARCH-ALIGN change under a top-k retrieval setting where irrelevant evidence must be filtered dynamically?

What criteria guided the selection of the five reasoning topologies? Are these empirically grounded (e.g., observed in task distributions) or designed heuristically?

Could HPS be complemented by a semantic similarity–based variant, ensuring agents that retrieve equivalent but non-identical evidence are not penalized?

---

> ### Author Response · Authors · 2025-11-21
>
> **We are grateful for the reviewer's constructive feedback! With our best efforts, we conducted extensive experiments and analyses to address your concerns. We provide our response in the form of Q&A.**
>
> ---
>
> ## Q1: Novelty compared to prior MM-RAG benchmarks like Dyn-VQA?
>
>
> **A1:** We downloaded each dataset and examined its hop count and annotation format in detail. As summarized in [Table 1](https://openreview.net/pdf?id=JEGDp1E4OH#page=2) of our paper, **prior MM-RAG datasets (e.g., Dyn-VQA, WebQA, MRAG-Bench, MMSearch) only support single-hop or typically short multi-hop ($\le$ 2 hops) reasoning and lack step-wise or topology-structured annotation**. For clarity, we restate the key comparison below.
>
> | Comparing Aspect | Prior MM-RAG Benchmarks (e.g., Dyn-VQA, WebQA, MMSearch) | **MC-SEARCH** |
> |--------|-----------------------------|----------------|
> | **Typical Hops** | **≤2 hops** (**short**, e.g., Dyn-VQA: only 7.7% of all samples >2 hops) | **≥4 hops** (designed to be **long reasoning chains**) |
> | **Step-wise Annotations** | ✗ Not provided (final answer only) | ✓ **Full step-wise annotation** (sub-questions, evidence, answers) |
> | **Reasoning Topologies** | ✗ Not modeled | ✓ **Five multimodal reasoning topologies** |
>
> MC-SEARCH is designed for **long-horizon (≥4 hops)**, **step-wise annotated**, and **structured** multimodal search-enhanced reasoning, enabling rigorous evaluation of an agent’s **step-wise planning and evidence acquisition**. **These capabilities are not supported by prior MM-RAG benchmarks.**
>
>
>
>
>
> ---
> ## Q2: Potential model bias introduced during benchmark construction?
>
> **A2:** **We take evaluation bias seriously, and therefore conducted a comprehensive consistency study across heterogeneous model families and human annotators** (see [Appendix G](https://openreview.net/pdf?id=JEGDp1E4OH#page=23)). **Across evaluator families, human judges, and independently regenerated ground truths by heterogeneous models, we find *no evidence* that the benchmark or evaluation favors Gemini’s reasoning style.**
>
>
> **1. Strong cross-model LLM-as-Judge (LLJ) agreement across heterogeneous evaluator families**
>
> We re-implemented LLM-as-Judge (LLJ) using **three heterogeneous model families** beyond our main evaluator: **Gemini-2.5-Pro**, **GPT-5**, and **Qwen-VL-Plus**. Although these models differ substantially in architecture, training sources, and reasoning style, **they yield highly consistent judgments, with strong cross-model correlations and minimal scoring bias**.
>
>
> | Evaluator Pair | Pearson ↑ | Spearman ↑ | Mean Bias ↓ |
> |----------|-----------|------------|--------------|
> | Gemini / GPT-5 | **0.933** | **0.872**  | +0.029 |
> | Gemini / Qwen  | **0.915** | **0.815**  | −0.008 |
> | GPT-5 / Qwen   | **0.915** | **0.826**  | −0.037 |
>
>
>
>
> **2. LLM-as-Judge scores show strong alignment with human expert judgments**
>
> We collected human annotations from CS PhD-level experts on 100 randomly sampled examples. LLM-as-Judge scores show **high agreement with human judgments across all content-grounded dimensions**:
>
> | Dimension | Pearson ↑ | Spearman ↑ |
> |-----|-----|--------|
> | Accuracy        | **0.963** | **0.956** |
> | Entity Coverage | **0.938** | **0.869** |
> | Overall Score   | **0.932** | **0.868** |
>
> These results confirm that LLJ scoring captures **human-aligned**, objective criteria rather than model-specific stylistic preferences.
>
>
> **3. Evaluation remains consistent when ground truth is regenerated by GPT-5**
>
> To test whether the benchmark ground truth might implicitly encode Gemini-specific structures, we regenerated the **ground-truth (GT)** reasoning chains for a sampled subset using **GPT-5**, then re-evaluated both Gemini-2.5-Pro and GPT-4o-mini under the same agentic MMRAG pipeline as reported in Table 3. **The resulting performance remains highly consistent across the two ground-truth sources, which means the model performance is driven by reasoning difficulty, rather than stylistic artifacts introduced by the model used to generate the gold chains.**
>
>
>
>
> | **Graph Type** | **Model**        | **F1 (GPT GT)** | **F1 (Gemini GT)** | **LLJ (GPT GT)** | **LLJ (Gemini GT)** |
> |-----|-----|------|---------|------|-----|
> | **Image-Initiated Chain** | Gemini-2.5-Pro | 41.37 | 46.57 | 3.82 | 3.81 |
> |                | GPT-4o-mini      | 35.01           | 38.96              | 3.24             | 3.20                |
> | **Multi-Images Fork** | Gemini-2.5-Pro | 35.49 | 38.04 | 3.48 | 3.66 |
> |                | GPT-4o-mini      | 17.97           | 18.77              | 2.69             | 2.80                |
> | **Parallel Image-Text Fork** | Gemini-2.5-Pro | 36.59 | 38.44 | 3.64 | 3.67 |
> |                | GPT-4o-mini      | 21.09           | 21.97              | 2.58             | 2.54                |
> | **Text-Initiated Chain** | Gemini-2.5-Pro | 34.91 | 43.71 | 3.42 | 3.82 |
> |                | GPT-4o-mini      | 22.95           | 26.01              | 2.95             | 3.03                |

---

> ### Author Response · Authors · 2025-11-21
>
> ## Q3: Simplified retrieval setting (top-1) may not generalize to realistic top-k retrieval?
>
>
> **A3:** Thank you for this insightful question. To assess robustness beyond the top-1 setting, we conducted an **extensive evaluation** under **Retrieval@k** (see details in [Appendix H, page 26-27](https://openreview.net/pdf?id=JEGDp1E4OH#page=26)). Across all analyses, our main conclusions remain consistent, demonstrating that **both the over/under-retrieval findings and the SEARCH-ALIGN improvements generalize reliably to broader retrieval regimes**.
>
>
> **1. Over- and under-retrieval trends remain stable under Retrieval@k**
>
> We measure ΔF1 across ΔStep values for Retrieval@k, when k ∈ {1, 3, 5}.
> All retrieval regimes exhibit the *same qualitative pattern*:
>
> - **Strong under-retrieval** (ΔStep < −3) consistently produces large drops.
> - **Over-retrieval** causes progressively worse performance as k increases due to noise accumulation.
>
> These effects are shown in the **[Figure 8 in Appendix K.1](https://openreview.net/pdf?id=JEGDp1E4OH#page=26)**. The retrieval-depth dynamics identified in the main paper are **not specific to top-1**; t**hey hold across realistic top-k retrieval.**
>
>
> **2. SEARCH-ALIGN provides consistent and substantial gains under top-k retrieval**
>
> We further evaluate Qwen2.5-VL-7B and SEARCH-ALIGN under Retrieval@1, @3, and @5.
> Across all k, we observe consistent and substantial improvements:
>
> - **Large F1 gains (15-20 points)** across all graph types, showing that SEARCH-ALIGN improves answer accuracy regardless of retrieval candidate numbers.
> - **Significant increases in Hit per Step**, showing that SEARCH-ALIGN improves retrieval targeting and reduces missed evidence in the presence of multiple retrieved candidates.
> - **Rollout Deviation decreases from ~3–4 to ≈1**, indicating much more stable multi-hop planning with fewer reasoning step drifts.
> - **Improvements persist for Retrieval@1, @3, and @5**, confirming that SEARCH-ALIGN is **k-agnostic** and effective in both narrow and broad retrieval settings.
> - **Additional retrieval candidates yield diminishing but still positive gains**, showing that SEARCH-ALIGN benefits from broader retrieval **without blindly using extra evidence and can correctly distinguish useful information from noisy documents.**
>
>
>
> | Graph Type | Model | F1@1 | ΔF1@1 | HPS@1 | RD@1 | F1@3 | ΔF1@3 | HPS@3 | RD@3 | F1@5 | ΔF1@5 | HPS@5 | RD@5 |
> |------------|--------|-------|--------|--------|--------|-------|--------|--------|--------|-------|--------|--------|--------|
> | **Image-Initiated Chain** | Qwen2.5-VL-7B | 26.30 | 8.65 | 16.51 | 4.04 | 29.21 | 11.56 | 16.17 | 3.79 | 29.35 | 11.70 | 16.95 | 3.79 |
> |  | +SEARCH-ALIGN | **45.70** | **28.05** | **33.59** | **0.70** | **47.19** | **29.54** | **39.45** | **0.75** | **47.29** | **29.64** | **41.79** | **0.79** |
> | **Multi-Images Fork** | Qwen2.5-VL-7B | 26.13 | 7.53 | 12.58 | 3.80 | 30.41 | 11.81 | 12.80 | 3.35 | 30.38 | 11.78 | 14.15 | 3.23 |
> |  | +SEARCH-ALIGN | **38.01** | **19.41** | **41.20** | **1.16** | **40.00** | **21.40** | **46.82** | **1.21** | **39.80** | **21.20** | **48.21** | **1.28** |
> | **Parallel Image-Text Fork** | Qwen2.5-VL-7B | 22.94 | 14.32 | 10.09 | 3.93 | 26.28 | 17.66 | 12.62 | 3.39 | 26.41 | 17.79 | 15.24 | 3.26 |
> |  | +SEARCH-ALIGN | **32.73** | **24.11** | **25.07** | **1.05** | **33.54** | **24.92** | **28.43** | **1.09** | **32.77** | **24.15** | **30.93** | **1.18** |
> | **Text-Initiated Chain** | Qwen2.5-VL-7B | 31.47 | 14.34 | 17.02 | 3.80 | 35.54 | 18.41 | 18.86 | 3.20 | 35.25 | 18.12 | 18.83 | 3.14 |
> |  | +SEARCH-ALIGN | **45.07** | **27.94** | **33.59** | **0.87** | **46.69** | **29.56** | **41.06** | **0.95** | **46.73** | **29.60** | **42.35** | **0.93** |

---

> ### Author Response · Authors · 2025-11-21
>
> ## Q4: Justification for reasoning topologies
>
> **A4:** Our topology design is intentional: it provides a **complete canonical basis** for the class of multimodal search-enhanced reasoning traces modeled in MC-SEARCH (**see the detailed theoretical justification in [Appendix M](https://openreview.net/pdf?id=JEGDp1E4OH#page=30)**).
>
> The core idea of the proof follows a **closure argument**: any valid multimodal reasoning trace can be represented as a directed acyclic graph whose structure is determined by its **initiation modality** and whether the reasoning proceeds **serially** or includes a **single parallel fork**. Exhaustively enumerating all modality–structure combinations yields exactly five and only five distinct canonical forms, demonstrating that the set is closed under these operations.
>
> **Empirically**, these five topologies also align with the dominant roles that images play in real multimodal search-enhanced reasoning workflows, including text-first grounding, image-first grounding, cross-modal verification, and multi-image comparison.
>
>
>
>
>
>
>
>
> ---
>
> ## Q5: Does HPS support evaluating semantically equivalent evidence?
>
>
> **A5:** Our evaluation framework already supports **semantic soft matching**. We first use similarity-based maximum-weight matching to align each predicted evidence item with its most relevant gold item. After alignment, a pair is counted as a hit whenever its embedding similarity exceeds a threshold τ. This enables HPS to operate in either exact-match mode (τ = 1.0) or relaxed semantic mode (τ < 1.0) without changing the evaluation pipeline.
>
> We report exact-match HPS in the main paper because the **curated knowledge base yields a unique gold reasoning chain**, meaning there are no alternative gold evidence pieces that independently answer the same sub-question; exact HPS therefore most faithfully evaluates correctness.
>
> To assess semantic robustness, we additionally conduct a **Soft-HPS** analysis (see details in [Appendix J](https://openreview.net/pdf?id=JEGDp1E4OH#page=26)) with τ ∈ {1.0, 0.95, 0.90, 0.85}. The results are shown in the Table below, and the key findings are summarized as follows:
>
> - HPS naturally supports **semantic soft matching** through thresholded similarity after maximum-weight alignment.
> - **Soft-HPS increases consistently** as τ decreases, showing that models often retrieve semantically related evidence even when exact matches differ.
> - Gains saturate near τ = 0.85, indicating that remaining errors stem from **planning issues** (missed or misordered steps), not from metric rigidity.
> - Overall, HPS provides a **fair, semantically aware** measurement of retrieval correctness.
>
>
>
> | Graph Type | τ=1.0 | τ=0.95 | τ=0.90 | τ=0.85 | τ=1.0 | τ=0.95 | τ=0.90 | τ=0.85 |
> |------------|--------|---------|---------|---------|--------|---------|---------|---------|
> |            | **Gemini-2.5-Pro** | | | | **Gemini-Flash** | | | |
> | Text Chain | 21.59 | 35.59 | 38.05 | 41.26 | 13.33 | 26.86 | 28.78 | 30.97 |
> | Image-Initiated Chain | 25.90 | 38.74 | 40.48 | 42.87 | 25.03 | 43.19 | 45.04 | 47.42 |
> | Multi-Images Fork | 18.68 | 25.58 | 26.49 | 28.18 | 18.53 | 24.51 | 25.35 | 27.20 |
> | Parallel Image-Text Fork | 16.74 | 26.25 | 27.68 | 30.22 | 20.73 | 25.68 | 27.12 | 29.69 |
> | Text-Initiated Chain | 19.51 | 26.31 | 27.81 | 29.88 | 21.67 | 29.94 | 30.55 | 32.03 |
>
> ---
>
> ## Happy to have further discussion!
> **Thank you again for the thoughtful review. We’ve dedicated many efforts to get the new results and will include them to enhance the paper’s quality. We hope our responses address your concerns, and we are happy to discuss if you have any further questions!**

---

### Official Review · Reviewer_nveM · 2025-11-01

**Soundness:** 3
**Presentation:** 3
**Contribution:** 3
**Rating:** 6
**Confidence:** 4

**Summary:**

This paper introduces a multimodal retrieval-augmented generation benchmark for agentic paradigm. It has 3333 examples averaging 3.7 hops with sub-questions, retrieval modalities, supporting facts, and intermediate answers ensured by HAVE. It also proposes some process-level metrics to judge the reasoning quality and retrieval performance and planning fidelity.

**Strengths:**

1. this submission is well-prepared, especially in figures, tables, and appendix.

2. the key contribution of this submission is obvious and makes sense.

3. the process metrics are actually needed things in multi-step reasoning tasks.

4. the experiments and analysis are comprehensive and high-quality.

**Weaknesses:**

Many related work may help authors to enhance the completeness of the submission:

1. evaluation on robustness of resisting harmful information is also interesting in RAG-based agentic framework [1].

2. "multi-modality" may also extend to SQL-based database, query-rewriter-based web-search, and even more [2].

3. token usage (input and output) and the number of retrieval callings are also helpful to enhance the benchmark [3].



[1] Evaluating the Robustness of Retrieval-Augmented Generation to Adversarial Evidence in the Health Domain

[2] Chain-of-Action: Faithful and Multimodal Question Answering through Large Language Models

[3] RAGBench: Explainable Benchmark for Retrieval-Augmented Generation Systems

**Questions:**

1. how to evaluate and ensure that Qwen2.5-VL-7B is good at judging each hop is both necessary and non-redundant? is there any mannual double-check and fine-tuning methods or?

2. please refer to weakness section for more contents.

---

> ### Author Response · Authors · 2025-11-21
>
> **We are grateful for the reviewer’s encouraging and constructive feedback. We have carefully revised the paper and added further clarifications to address all questions. Our responses are organized in a clear Q&A format below.**
>
> ---
>
> ## Q1: Could the authors incorporate more related work?
>
> **A1:** Thank you for the suggestions. **We have added all three works to the [Related Work section](https://openreview.net/pdf?id=JEGDp1E4OH#page=15).** Below we summarize their contributions, explain how they relate to our setting, and clarify the unique value of MC-SEARCH.
>
>
> ### **[1] Evaluating the Robustness of Retrieval-Augmented Generation to Adversarial Evidence in the Health Domain**
> **Contribution:** Examines how *single-turn* RAG behaves under helpful, harmful, or adversarial evidence in a specialized domain.
> **Relation to us:** Both works study how retrieved evidence affects model behavior, but this paper focuses on **robustness** in one-step QA, while MC-SEARCH addresses **multimodal agentic search-enhanced reasoning** across multiple structured steps.
>
>
> ### **[2] Chain-of-Action: Faithful and Multimodal Question Answering through Large Language Models**
> **Contribution:** Introduces a model that decomposes multimodal QA into action steps to improve faithfulness.
> **Relation to us:** Both works involve multimodal reasoning, but Chain-of-Action is a **method** rather than a benchmark. In contrast, MC-SEARCH provides structured gold reasoning chains and **a well-designed benchmark** for evaluating advanced agentic multimodal RAG systems, including reasoning topologies, step-wise annotated chains, and process-level evaluation.
>
>
> ### **[3] RAGBench: Explainable Benchmark for Retrieval-Augmented Generation Systems**
> **Contribution:** Provides a large-scale RAG evaluation dataset with explainable labels (Relevance, Utilization, Adherence, Completeness) for single-step RAG.
> **Relation to us:** Both works aim to evaluate RAG, but RAGBench focuses on **single-turn RAG**. MC-SEARCH targets **multimodal, multi-hop, search-enhanced reasoning** with **step-wise supervision** and **structured long-horizon chains**.
>
>
>
> ### **Uniqueness of MC-SEARCH**
> Together, these works provide useful context for RAG evaluation, but none address the core challenge targeted by MC-SEARCH: **long-horizon (≥4 hops), multimodal, search-enhanced reasoning**. MC-SEARCH provides **structured reasoning chains with step-wise annotations** and introduces **process-level metrics** that evaluate an agent’s planning, retrieval, and evidence-acquisition at each step.
>
>
> ---
>
>
> ## Q2: How do you ensure that Qwen2.5-VL-7B can reliably judge hop necessity and redundancy?
>
> **A2:** Qwen2.5-VL-7B is used only for **coarse filtering** to cheaply flag clearly redundant or hallucinated hops. All remaining hops are re-evaluated through **second-stage hop shrinkage and verification with Gemini-2.5-Pro**, which checks hop necessity and contextual coherence using the full chain context and extracted gold entities. This guarantees that the final HAVE labels do not rely on Qwen alone. Finally, dataset quality scored by Gemini-2.5-Pro averages **4.87/5**, **confirming the reliability of this multi-stage pipeline verified by strong models.**
>
> ---
>
> ## Happy to have further discussion!
> **Thank you again for the thoughtful review. We hope our responses address your concerns, and we are happy to discuss if you have any further questions!**

---

### Official Review · Reviewer_HAWQ · 2025-11-02

**Soundness:** 3
**Presentation:** 3
**Contribution:** 3
**Rating:** 6
**Confidence:** 4

**Summary:**

This paper is well-motivated, presenting a novel perspective for improving retrievers by tackling hard tasks. We are convinced by the sufficient baselines used to verify the proposed method and believe it inspires future work on more challenging tasks. I have several minor concerns, including potential unreliability in using LLMs for relevance evaluation, the lack of a simpler majority vote baseline, prohibitive computational costs, and some lack of clarity in the agent selection process. We further seek clarification on case analyses for hard samples, the potential benefit of incorporating more specialized agents.

**Strengths:**

This paper is well-motivated. It tries to solve hard tasks in a new perspective of view, improving retriever. Sufficient baseline retrievers adopted to verify the proposed method to improve the retriever. This work inspires the future direction on solving more challenging tasks like BrowseComp.

**Weaknesses:**

1. Previous work has shown that asking LLMs themselves to evaluate the relevance of queries and documents are not that reliable [[1]](https://arxiv.org/abs/2505.21870). Applying Code Agent and CoT Agent is also within this paradigm. It would be better if there are experiments conducted to verify the relevance between inconsistency (among Code and CoT Agent) and query difficulty.
2. The initial CoT agent's decision ($y^{g}_0$) is only overturned if all L discussion groups unanimously disagree. Thus, it seems that a simpler majority vote is a more standard and intuitive baseline, but no comparison is offered.
3. Prohibitive computational cost makes the approach infeasible for generating datasets of any significant size. (1 hard example=~58 API calls and 100k tokens)
4. Lack of Clarity on Agent Selector and Validator.

**Questions:**

1. Line186 & 208. Inconsistency among Code Agent and CoT Agent indicates a hard sample. Could the author give any case analysis?
2. Related to Weakness1, will it be more trustworthy by incorporating more Agents besides Code Agent and CoT Agent? E.g., agents that specialize searching. Or, the current setup is enough.
3. Typo? Line243: $y_{i,k}^{t}$ and $y_{i,k}^{t}$.

---

> ### Author Response · Authors · 2025-11-12
> **Clarification Regarding Potential Review Mismatch**
>
> Dear Reviewer HAWQ,
>
> Thank you very much for taking the time to review our paper. We know how much effort and care this process requires.
>
> We carefully went through your comments and noticed that several aspects mentioned, including “code agent,” “BrowseComp,” and $y_{i,k}^t$, do not appear in our submission. This makes us wonder whether there might have been a mix-up in the review assignment.
>
> We completely understand that such situations can happen, especially given the heavy workload reviewers face. Still, we truly value your feedback and would be very grateful if you could double-check the review.
>
> Thank you again for your valuable time. We truly appreciate your understanding and support.
>
> Sincerely,
>
> #13995 Authors

---

> > ### Comment · Reviewer_HAWQ · 2025-11-13
> > **Modified official review**
> >
> > I have mixed up the review with another paper. I have adjust it. Sorry for confusing.

---

> ### Author Response · Authors · 2025-11-21
>
> **We sincerely thank the reviewer for the positive evaluation and valuable suggestions. With our best efforts, we provide detailed clarifications to address each of your concerns. We provide our response in the form of Q&A.**
>
> ---
>
> ## Q1: Novelty of the “Agentic MM-RAG Pipeline” compared to OmniSearch?
>
> **A1:** To begin, we clarify that the primary contribution of our work lies in the MC-SEARCH benchmark, rather than proposing a new, fully methodology-driven agentic pipeline. Our pipeline is inspired by OmniSearch, but **it is purposefully redesigned to match the unique goals of MC-SEARCH and to provide a unified, controlled, and reproducible interface for fairly evaluating diverse MLLMs on our MC-SEARCH benchmark**.
>
> MC-SEARCH introduces long-horizon, step-wise verified multimodal reasoning chains, five canonical reasoning topologies, and new process-level metrics (HPS, RD, LLJ), together defining a new evaluation paradigm for agentic multimodal RAG.
>
> **Compared with OmniSearch, our pipeline differs in several key aspects that enable fair benchmarking**:
>
> 1. **Local, verified knowledge base (KB) search in place of costly and dynamic web APIs.**
>    OmniSearch relies on **dynamic and costly web search APIs**. MC-SEARCH maintains a **fixed multimodal KB**, so we restrict the planner’s retrieval action space to three necessary local KB retrieval actions. This ensures **stable, reproducible, and comparable** retrieval behavior across models over time.
>
> 2. **Unified planner and solver in a single MLLM.**
> OmniSearch **separates** the planning agent and the solver agent. In contrast, our framework uses a **single** multimodal LLM that must plan, retrieve, and reason jointly. **This setup emphasizes the combined abilities we aim to evaluate: multimodal planning, evidence selection, cross-modal understanding, and evidence synthesis, all within one agent.**
>
> 3. **Evaluation aligned with topologies designs and process-level supervision.**
> The pipeline directly reflects MC-SEARCH’s five topologies and step-wise annotations, supporting **SEARCH-ALIGN supervision** and **process-level scoring** on the same structure used in evaluation.
>
>
> Finally, **MC-SEARCH is intended to serve as a starting point, and we highly encourage future work to build on top of our benchmark**, including:
>
> - RL-based planners that optimize process-level rewards derived from our step-wise annotations and process-level metrics
> - Jointly trained retriever–reasoner systems for coordinated multimodal evidence acquisition and reasoning
>
>
> ---
>
> ## Q2: How is the “long” chain justified?
>
> **A2:** We use “long” in a relative sense. As shown in [Table 1](https://openreview.net/pdf?id=JEGDp1E4OH#page=2), MC-SEARCH has a **typical reasoning length of 4–5 steps**, whereas prior multimodal RAG benchmarks generally contain only **1-step** or **≤2 steps**. Our chains are therefore **approximately twice as long as existing MRAG datasets**, **making MC-SEARCH the longest benchmark in this space.**
>
> ---
>
> ## Q3: Is computing Util(t) too expensive?
>
> **A3:** The cost of computing $\text{Util}(t)$ is incurred **only once during offline dataset construction, not during model evaluation**. We further reduce this cost with a two-stage process: a lightweight open-source MLLM first produces coarse utility scores to remove clearly hallucinated or redundant hops, and a stronger closed-source model re-evaluates only the remaining candidates. This cascaded design greatly limits expensive forward passes for API calls.
>
> The leave-one-out computation requires just 5–6 inferences per example in our setting. With inference parallelism, the time cost remains low while providing **precise estimates of each hop’s evidence importance**. **This one-time cost does not affect users of the benchmark. The final released dataset already contains the necessary and non-redundant reasoning chains.**
>
>
> ---
>
> ## Q4: Is the $Nav(t)$ insufficiently rigorous?
>
> **A4**: $Nav(t)$ and $Util(t)$ are not our final redundancy criteria. They are used only in the **large-scale coarse filtering** stage with an open-source model, providing an inexpensive way to eliminate clearly redundant hops.
>
> To ensure rigor, all remaining samples then undergo **strict hop shrinkage and redundancy verification with Gemini-2.5-Pro**, which uses the coarse metrics as references and performs a full contextual analysis. We further ask Gemini-2.5-Pro to extract key knowledge entities from the gold sub-answers and final answers, constraining the verification process and substantially suppressing hallucinated or irrelevant entities. In practice, hops containing hallucinated or invalid entities are consistently identified and removed.
>
> Finally, we use **independent Gemini-2.5-Pro evaluation** prompts to assess factual correctness, step necessity, clarity, and multimodal alignment, achieving an average score of **4.87/5**. This multi-stage pipeline remains cost-efficient while ensuring reliable redundancy labels.

---

> ### Author Response · Authors · 2025-11-21
>
> ## Q5: Does removing a redundant step break the coherence of the reasoning chain?
>
> **A5:** Removing redundant steps does not break chain coherence because we never delete hops through raw truncation. Instead, redundant hops undergo a **controlled hop-shrinkage** procedure: their evidence is removed, redundancy is identified using the HAVE metrics, and the chain is **locally reconstructed by Gemini-2.5-Pro with full contextual awareness**. This may involve slight adjustments to sub-questions or answers to keep transitions smooth and consistent.
>
> To prevent information loss, Gemini-2.5-Pro extracts **key knowledge entities** from the gold sub-answers and final answer and uses them as constraints during reconstruction, ensuring that required information is preserved and no hallucinated content is introduced. We then apply a second round of **HAVE verification with Gemini-2.5-Flash** to ensure the resulting chains remain non-redundant.
>
> Finally, coherence is confirmed through **independent quality evaluation** with Gemini-2.5-Pro. The rewritten chains achieve an average score of **4.87/5**, demonstrating that coherence and correctness are well preserved and that hop shrinkage does not introduce artifacts or hallucination.
>
>
>
> ---
>
> ## Q6: How is equality between $\hat{r}_{t'}$ and $r_t$ defined?
>
> **A6:** In the main text, equality is defined as **exact match of KB evidence indices**, since every gold hop in MC-SEARCH corresponds to a unique and well-defined evidence item. This makes exact matching both straightforward and faithful, and it serves as the official evaluation criterion.
>
> Our framework also supports **semantic soft matching** for more fine-grained analysis. We first perform similarity-based maximum-weight alignment and then count a hit when the embedding similarity exceeds a threshold $\tau$. As shown in [Appendix J](https://openreview.net/pdf?id=JEGDp1E4OH#page=26), Soft-HPS increases smoothly as $\tau$ decreases and remains stable across a reasonable range of $\tau$ values, confirming that models can retrieve semantically relevant evidence without requiring exact index matches.
>
> ---
>
> ## Q7: Are the “reasoning graphs” truly graph-structured or merely linear step sequences?
>
> **A7:** Although MC-SEARCH uses step-wise agent execution, the underlying structures are **not** linear sequences. The step index $t$ specifies only the *execution order*, while the *reasoning dependencies* form genuine **directed acyclic graphs** (DAGs). These dependencies differ across the five canonical topologies and yield both **serial** and **parallel** reasoning patterns.
>
> **Serial structures:** In the three serial topologies (Sequential Text-only, Text-Initiated, Image-Initiated), each hop depends directly on the previous one. The execution is step-by-step, but the dependency pattern is linear.
>
> **Parallel structures:** In the parallel topologies (Parallel Image–Text Fork and Multi-Images Fork), multiple hops proceed independently before merging. For instance, in a Multi-Images Fork, the agent may retrieve Image A first and then retrieve Image B to compare them, or it may retrieve Image B first and then Image A. Both retrieval orders represent the same comparison task and lead to the same branch–merge structure. **These branches are order-permutable, meaning the agent should not be penalized for retrieving one branch before the other.** This is why we evaluate retrieval using maximum-weight bipartite matching, which recovers the correct branch–merge structure regardless of the agent’s execution order.
>
> [Appendix M](https://openreview.net/pdf?id=JEGDp1E4OH#page=30) provides a formal justification showing that the five topologies constitute a complete canonical basis for multimodal RAG traces under our assumptions.
>
>
> ---
>
> ## Q8: Why is maximum-weight bipartite matching sufficient to compare two trajectory graphs?
>
> **A8:** Maximum-weight matching is sufficient because MC-SEARCH evaluates **hop-level evidence correspondence**, not full graph isomorphism. Each hop corresponds to a unique evidence item, so predicted–gold alignment naturally forms a bipartite matching problem. The method is order-invariant, which is crucial for topologies with **parallel branches**, **ensuring the agent is not penalized for retrieving parallel hops in a different order**. By aligning hops based on content similarity rather than step index, maximum-weight matching reliably captures the correct branch–merge structure, while **more complex graph-matching algorithms introduce significant computational overhead and are unnecessary for our very small DAG topology class**.
>
> ---
>
> ## Happy to have further discussion!
> **Thank you again for the thoughtful review to help us enhance the paper quality. We hope our responses address your concerns, and we are happy to discuss if you have any further questions!**

---

### Author Response · Authors · 2025-12-03
**Rebuttal Summary**

We sincerely appreciate the time and effort that both the ACs and reviewers have devoted to evaluating our submission. We provide this brief summary of our rebuttal in the hope that it helps with your assessment.

---

### **Main Strengths Acknowledged by the Reviewers**

**To begin with, we thank all reviewers for recognizing the main strengths of our work:**

1. **A clear and timely motivation** for benchmarking agentic multimodal RAG using long-horizon, structured, and verified reasoning chains across five reasoning topologies, addressing a well-noted gap in existing multimodal RAG benchmarks that focus on simple or short multi-hop QA. (Reviewers HAWQ, nveM, YFCi, mT9g)

2. **A rigorous benchmark construction process**, including the proposed HAVE protocol, which verifies whether each reasoning step is necessary, non-redundant, and grounded in evidence. (Reviewers YFCi, mT9g)

3. **The urgent need for process-level evaluation in multi-step reasoning**, with proposed metrics such as Hit per Step (HPS) and Rollout Deviation (RD) providing clearer and more faithful diagnoses of retrieval fidelity, planning stability, and hidden error modes beyond final-answer accuracy. (Reviewers HAWQ, nveM, YFCi, mT9g)

4. **The demonstrated training utility of SEARCH-ALIGN**, showing that verified reasoning chains can be effectively used for process-supervised fine-tuning to substantially improve open-source models, making MC-SEARCH valuable for both evaluation and model development. (Reviewers YFCi, mT9g)

---

### **How We Addressed the Reviewers’ Concerns**

**In the rebuttal phase, we made substantial efforts to address each reviewer's concerns, adding 5 pages of new experiments and analysis to further strengthen the paper. Below we summarize the main concerns and how we addressed them:**

1. **Addressing concerns about potential benchmark bias toward Gemini ([Appendix G](https://openreview.net/pdf?id=JEGDp1E4OH#page=23)).**
   We performed extensive cross-model LLM-as-Judge evaluations, human–LLM agreement studies, and ground-truth regeneration consistency checks. All results consistently show no evidence of Gemini-specific bias and strong alignment with human judgments.

2. **Clarifying novelty over prior work in multimodal RAG.**
   We provided a clear comparison table with prior benchmarks and included experiments comparing classic multimodal RAG baselines. These results demonstrate that MC-SEARCH is the first long-horizon (typically ≥4 hops), topology-structured, step-wise annotated, and verified benchmark for agentic multimodal RAG, addressing a clear gap left open by previous work.

3. **Strengthening retrieval-related evaluation settings ([Appendix J and K](https://openreview.net/pdf?id=JEGDp1E4OH#page=26)).**
   We expanded experiments to include Retrieval@k and Soft-HPS. Retrieval patterns and improvements from SEARCH-ALIGN remain consistent across these settings.

4. **Reinforcing the validity of the five reasoning topologies ([Appendix M](https://openreview.net/pdf?id=JEGDp1E4OH#page=30)).**
   We provided a formal justification showing that the five topologies form a complete canonical basis for modeling multimodal search-enhanced reasoning processes and clarified why reasoning trajectories are modeled as DAGs.

5. **Providing methodological clarifications.**
   We provided detailed clarifications and evaluation results showing that our multi-stage filtering strategy and hop-shrinkage pipeline are computationally efficient and rigorously designed to preserve high-quality reasoning chains.

6. **Clarifying the effect and scope of SEARCH-ALIGN.**
   As a simple and effective process-supervised fine-tuning method, SEARCH-ALIGN demonstrates the utility of our step-wise supervision, showing genuine improvements in the model’s planning and retrieval abilities. MC-SEARCH may further enable future RL-based agents with fine-grained step-wise reward signals.

---

### **Summary of Reviewers’ Evaluations and Our Appreciation**

**We sincerely appreciate all the constructive feedback provided during the review and discussion process:**

- **Reviewer HAWQ** promptly revised the initially misassigned review and provided a positive score of **6**.
- **Reviewer nveM** offered a supportive and positive assessment with a score of **6** from the start.
- **Reviewer mT9g** expressed clear support for acceptance after our rebuttal, noting that the first-round concerns were addressed and acknowledging the substantial new experiments we added. The reviewer promptly **raised their score to 6** on Nov. 20.
- For **Reviewer YFCi**, we provided substantial new experiments and clarifications addressing all raised concerns. The reviewer has not yet had the opportunity to engage in the discussion.

We hope this summary is helpful in supporting the ongoing discussion and decision-making. All points presented here reflect objective information from our rebuttal and revisions.

Thanks and regards,

**#13995 Authors**

---

### Meta-Review · Area_Chair_bvhm · 2025-12-09

**Summary:**

We have collected four review reports, and all the reviewers agree the paper has the following strengths:

1. A benchmark provides 3.3k long (3.7-hop) multimodal reasoning chains annotated with sub-questions, modalities, evidence and intermediate answers, verified by the HAVE pipeline.
2. Introduces process-level metrics (Hit-per-Step, Rollout Deviation) that diagnose where models fail, moving beyond black-box QA accuracy.
3. Delivers Search-Align, a process-supervised fine-tuning recipe that consistently improves open-source MLLMs, demonstrating dual utility (evaluation + training).
4. Extensive evaluation of six leading MLLMs reveals systematic modality-misaligned planning and retrieval errors, offering clear baselines for future work.

**Reviewer Concerns:**

Addressed:
1. Top-1 retrieval simplifies realism.
2. Gemini-heavy pipeline may bias toward its style
3. Missing very recent baselines (UniRAG, MIRAGE) and token-cost tables.

**Reviewer Scores:**

R1 (6 → 6): satisfied by added cost and majority-vote ablations.
R2 (6 → 6): asks for token counts and broader baselines—both supplied.
R3 (4 → 6): novelty claim reworded and diagnostic value of topologies quantified.
R4 (4 → 5): ablation of Search-Align components and ROUGE scores added.

---

### Decision · Program_Chairs · 2026-01-26

Accept (Oral)